EMBO
Molecular Medicine

# Therapeutic development of group B *Streptococcus* meningitis by targeting a host cell signaling network involving EGFR

Ningyu Zhu[1], Chengxian Zhang[1], Atish Prakash[1], Zheng Hou[1], Wei Liu[1], Weifeng She[1], Andrew Morris[2] & Kwang Sik Kim[1,*] (iD)

## Abstract

Group B *Streptococcus* (GBS) remains the most common Gram-positive bacterium causing neonatal meningitis and GBS meningitis continues to be an important cause of mortality and morbidity. In this study, we showed that GBS penetration into the brain occurred initially in the meningeal and cortex capillaries, and exploits a defined host cell signaling network comprised of S1P$_2$, EGFR, and CysLT1. GBS exploitation of such network in penetration of the blood–brain barrier was demonstrated by targeting S1P$_2$, EGFR, and CysLT1 using pharmacological inhibition, gene knockout and knockdown cells, and gene knockout animals, as well as interrogation of the network (up- and downstream of each other). More importantly, counteracting such targets as a therapeutic adjunct to antibiotic therapy was beneficial in improving the outcome of animals with GBS meningitis. These findings indicate that investigating GBS penetration of the blood–brain barrier provides a novel approach for therapeutic development of GBS meningitis.

**Keywords** blood–brain barrier; CysLTs; EGFR; GBS meningitis; S1P
**Subject Categories** Microbiology, Virology & Host Pathogen Interaction; Neuroscience

## Introduction

Group B *Streptococcus* (GBS) remains the most common Gram-positive bacterium causing neonatal meningitis and GBS meningitis continues to be an important cause of mortality and morbidity (Kim, 2008, 2010; Romain *et al*, 2018; O'Sullivan *et al*, 2019). Group B *Streptococcus* strains are uniformly susceptible to penicillins, and poor outcome of GBS meningitis is not due to emergence of non-susceptible GBS strains. These findings indicate new approaches are needed for development of improved therapy for GBS meningitis. A major limiting factor for discovery of new therapeutic targets is our incomplete understanding of the pathogenesis of GBS meningitis (Kim, 2008; Maisey *et al*, 2008).

Several lines of evidence from human cases and experimental animal models of GBS meningitis suggest that GBS penetration into the brain occurs in the cerebral microvessels (Berman & Banker, 1966; Ferrieri *et al*, 1980; Doran *et al*, 2005; Tazi *et al*, 2010), but it remains incompletely understood how GBS penetrates the blood–brain barrier. Meningitis-causing pathogens penetrate the blood–brain barrier via transcellular, paracellular and/or hijacking infected phagocytes, so-called Trojan-horse mechanisms (Kim, 2008). Transcellular mechanism is defined as microbial penetration through barrier cells without exhibiting any microbes between the cells and/or disruption of intercellular tight junction, while paracellular mechanism refers to microbial penetration between barrier cells with and/or without demonstration of tight junction disruption (Kim, 2008). Currently, controversy exists on whether GBS exploits transcellular and/or paracellular penetration of the blood–brain barrier (Doran *et al*, 2016).

Group B *Streptococcus* penetration into the brain has been shown to occur without affecting the blood–brain barrier permeability and without accompanying immune cells (Kim *et al*, 1997), suggesting that GBS invasion of the brain is likely to exploit a transcellular penetration of the blood–brain barrier. This concept was shown in the current study, where GBS penetration occurred initially in the meningeal and cortex capillaries, and there was no leakage of intravascular small molecule tracer (sulfo-NHS-biotin with a molecular weight of 443 Da) and no recruitment of immune cells to the sites of GBS penetration.

Since GBS penetrates into the cerebral microvasculature, we used the *in vitro* blood–brain barrier model with human brain microvascular endothelial cells (HBMECs) to investigate how GBS penetrates the blood–brain barrier. Our HBMEC monolayer upon cultivation on collagen-coated Transwells exhibits spatial organization of tight and adherens junction proteins as well as a polarized monolayer, a unique property of the blood–brain barrier endothelial cells (Stins *et al*, 1997, 2001; Nizet *et al*, 1997; Rüffer *et al*, 2004; Kim *et al*, 2004). We showed that GBS invasion of HBMEC monolayer

---

1 Division of Pediatric Infectious Diseases, Johns Hopkins University School of Medicine, Baltimore, MD, USA
2 Division of Cardiovascular Medicine, The Gill Heart Institute, University of Kentucky, Lexington, KY, USA
*Corresponding author. Tel: +1 410 6140058; E-mail: kwangkim@jhmi.edu

occurred without affecting the integrity of HBMEC monolayer, as assessed by live/dead staining (Molecular Probes) as well as measurement of transendothelial electrical resistance (TEER) before and after infection (Nizet *et al*, 1997; Maruvada *et al*, 2011). These *in vitro* and *in vivo* findings demonstrate that GBS invasion of the brain is likely to occur via a transcellular penetration of the blood–brain barrier and less likely to exploit paracellular and Trojan-horse mechanisms.

We hypothesize that elucidation of GBS invasion of the blood–brain barrier will enhance our knowledge on the pathogenesis of GBS meningitis. We carried out transcriptome analysis of HBMEC (RNA-seq) in response to GBS infection. The network analysis of the RNA-seq data revealed that the epidermal growth factor receptor (EGFR) pathway is activated during GBS infection. The contribution of EGFR to GBS invasion of the blood–brain barrier is supported by our demonstration that gefitinib, a FDA-approved inhibitor of EGFR (Wakeling *et al*, 2002), significantly inhibited GBS invasion of HBMEC monolayer as well as penetration into the brain. We elucidated host cell signal transduction pathways involving EGFR that contributed to GBS penetration of the blood–brain barrier. We showed that sphingosine 1-phosphate (S1P)-S1P$_2$ represents upstream molecules of EGFR, while cytosolic phospholipase $_2\alpha$ (cPLA$_2\alpha$), cysteinyl leukotrienes (CysLTs), and ezrin–radixin–moesin (ERM) represent downstream molecules of EGFR and that S1P$_2$-EGFR-CysLTs formed a defined host cell signaling network exploited by GBS for penetration of the blood–brain barrier *in vitro* and *in vivo*. More importantly, counteracting such network as a therapeutic adjunct to antibiotic therapy improved the outcome of animals with GBS meningitis.

This is the first report that GBS exploits a defined host cell signaling network comprised of specific host factors (S1P$_2$, EGFR, and CysLT1) for penetration of the blood–brain barrier and that counteracting such network represents a new approach for development of therapeutic targets for GBS meningitis.

## Results

### RNA-seq analysis of HBMEC in response to GBS infection

To investigate GBS interaction with the blood–brain barrier, we performed the HBMEC transcriptional responses to GBS infection for 2 and 6 h and determined upstream regulator analysis on the sets of host genes that were differentially expressed with and without infection ($P < 0.05$) (Dataset EV1; Liu *et al*, 2015; Watkins *et al*, 2018) by the Ingenuity Pathway Analysis (IPA) software (Ingenuity Systems; http://www.ingenuity.com). The raw sequencing reads have been submitted to the NCBI sequence read archive (SRA) under BioProject accession no. PRJNA632824. Analysis of differentially expressed genes predicted the modulation of a total of 178 different signaling proteins from HBMEC infected with GBS during at least one time point of the infection, including 132 activated signaling proteins and 46 repressed signaling proteins. We then performed a protein–protein interaction enrichment analysis using Metascape (http://metascape.org; Li *et al*, 2018; Zhou *et al*, 2019) and identified six protein–protein interaction networks (Fig EV1). The significantly up-regulated EGFR is the core of one protein network, suggesting that EGFR plays a role in GBS infection of HBMEC (Fig 1A).

### GBS penetration into the brain without affecting the functional and structural alterations of the blood–brain barrier

In order to study the role of EGFR in GBS penetration into the brain, we used the mouse model of experimental hematogenous GBS meningitis. We showed that GBS penetration occurred initially in the meningeal and cortex capillaries. This was shown by the demonstration of intravenously administered GBS in the meningeal and cortex capillaries at 1 h following bacterial inoculation. A few bacteria were found outside the capillaries of the meninges and cortex

**Figure 1.  Group B *Streptococcus* (GBS) exploits host EGFR for penetration of the blood–brain barrier.**

A   Protein–Protein interaction enrichment analysis of differentially expressed host factors in HBMEC with and without GBS infection revealing signaling network involving EGFR.

B   Wild-type mice received strain GFP-K79 via the tail vein and 1 h later, animals were perfused and the meninges, cortex and choroid plexus were obtained for demonstration of intravenously injected bacteria. The arrows represent bacteria co-localized with capillaries (e.g., within capillaries), and arrowheads represent bacteria outside the capillaries (e.g., exited from the capillaries). A few bacteria were successfully passed through the meningeal and cortex capillaries, but no bacteria were demonstrated in the choroid plexus at this time. Scale bar = 100 μm. The completed figures with control and 12 h infection group were shown in Fig EV2A–C.

C   Sulfo-NHS-biotin was administered via intraperitoneal injection 10 min before perfusion for assessing the blood–brain barrier permeability, i.e., extravasation is indicative of leakage from the intravascular lumen. Sulfo-NHS-Biotin was confined to the capillaries of the meninges and cortex (as shown with arrows), and there was no evidence of extravasation. Scale bar = 100 μm. The completed figures with control and 12 h infection group were shown in Fig EV2D and E.

D   Relative invasion frequency of GBS strain K79 in HBMEC with or without EGFR inhibitor (gefitinib). Data represent the means ± SEM from three independent experiments, with statistical analysis by Student's *t*-test, **$P = 0.0020$, ***$P = 8.6E-06$ (from left to right).

E   Bacterial counts recovered from the blood and brain in wild-type mice receiving vehicle control ($n = 5$) or gefitinib (10 mg/kg) ($n = 6$), infected with strain K79 for 1 h. Data represent the means ± SEM with Wilcoxon rank sum test, **$P = 0.0038$.

F   Tyrosine phosphorylation of EGFR in HBMEC in response to GBS strain K79 infection at 15-min and 60-min post-infection.

G   EGFR protein expression in EGFR knockout HBMEC using CRISPR/Cas9 (left panel) and relative invasion frequency of 8 meningitis isolates of GBS strains in EGFR knockout and control HBMEC (right panel). Data represent the means ± SEM from three independent experiments, with statistical analysis by Student's *t*-test. From left to right, $P$ value of K79 is 0.0059, $P$ value of K160 is 0.00024, $P$ value of K161 is 0.00080, $P$ value of K181 is 0.00042, $P$ value of K226 is 0.00011, $P$ value of K237 is 0.00072, $P$ value of K238 is 0.0093, and $P$ value of P539 is 0.00013 (right panel).

H   GBS strain K79 traversal of HBMEC monolayer was significantly decreased in EGFR knockout HBMEC compare to control HBMEC. Data represent the means ± SEM from three independent experiments, with statistical analysis by Student's *t*-test, **$P = 0.0027$.

I   Bacterial counts recovered from the blood and brain of EGFR conditional knockout ($n = 5$) and control mice ($n = 5$) 1 h after intravenous inoculation with strain K79. Data represent the means ± SEM with statistical analysis by Student's t-test, ***$P = 0.00026$.

Source data are available online for this figure.

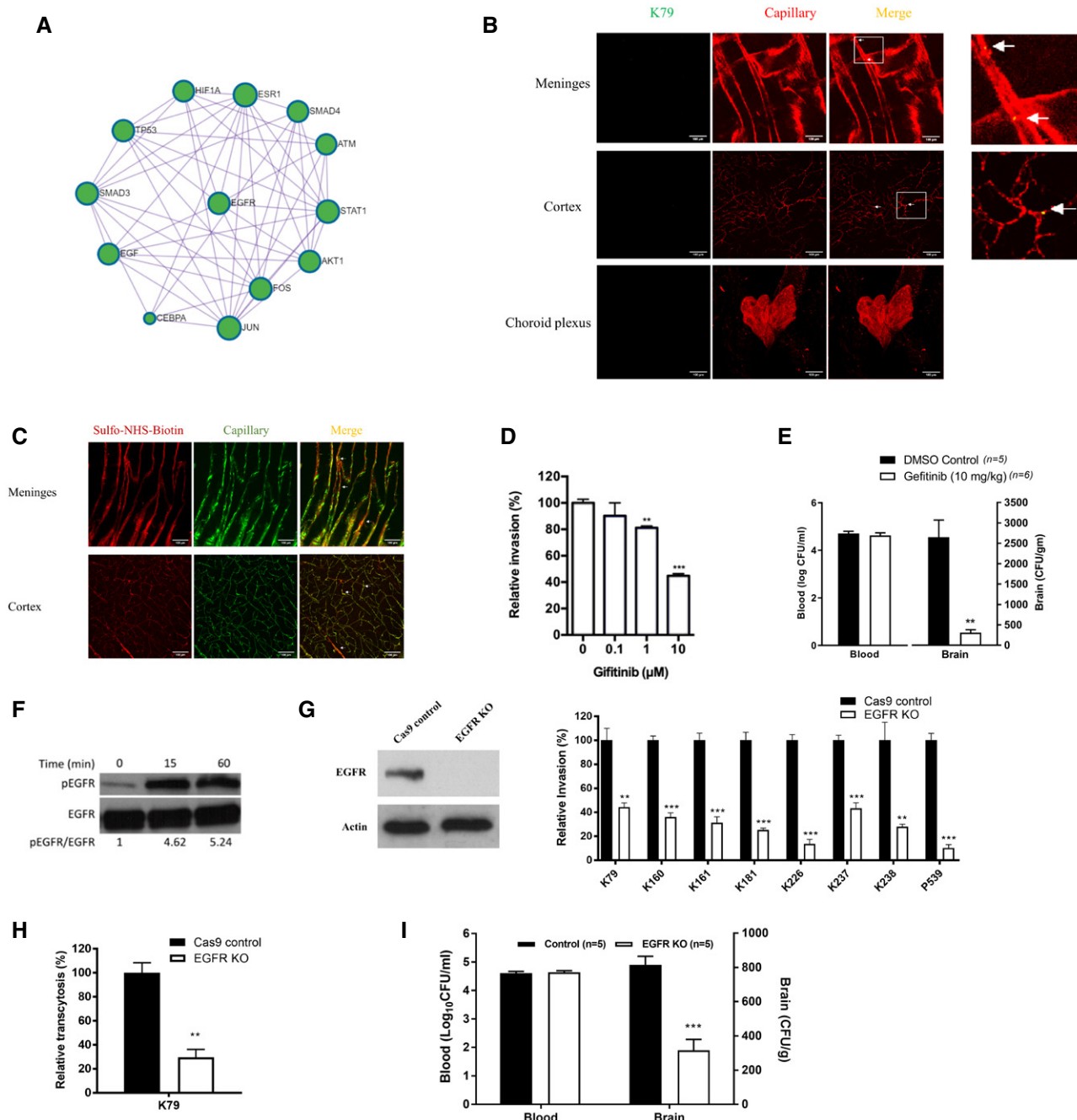

Figure 1.

(Figs 1B and EV2), indicating successful penetration of GBS into the brain at this time, while there was no demonstration of GBS in the choroid plexus (Figs 1B and EV2). These findings demonstrate that the meningeal and cortex capillaries, not the choroid plexus, are the portal of entry for circulating GBS penetration into the brain, but more GBS were found to be present in the meningeal capillaries, indicating that GBS invasion involved initially meningeal and subsequently cortex capillaries. It is important to note that intravascular small molecule tracer (i.e., Sulfo-NHS-biotin with m.w. of 443 Da) was confined to the meningeal and cortex capillaries, and there was no extravasation (Figs 1C and EV2). We also examined the

structural integrity of the blood–brain barrier during the GBS penetration of the blood–brain barrier, by assessing claudin-5 staining to visualize the tight junction of the barrier endothelial cells. The claudin-5 staining one hour after GBS administration via the tail vein demonstrated GBS in the brain capillaries and successful penetration into the brain, and there was no disruption of the claudin-5 continuity (Fig EV3A). Taken together, these findings indicate that GBS penetration into the brain occurs in the meningeal and cortex capillaries without affecting the blood–brain barrier permeability even to a small molecule and without disruption of the blood–brain barrier structural integrity, demonstrating that GBS exploits a

transcellular penetration of the blood–brain barrier, and supporting the use of HBMEC monolayer for investigating the pathogenesis of GBS meningitis.

## GBS exploits EGFR in penetration of the blood–brain barrier

The network analysis of RNA-seq data suggested the role of EGFR in GBS infection of HBMEC monolayer, and we examined gefitinib, a FDA-approved selective inhibitor of EGFR, for its effect on GBS invasion of the blood–brain barrier *in vitro* and *in vivo*. We showed that gefitinib inhibited GBS invasion of HBMEC monolayer in a dose-dependent manner (Fig 1D). We also examined the effect of gefitinib in GBS penetration into the brain by intraperitoneal administration 1 h before intravenous challenge of bacteria and determining bacterial counts recovered from the brain. Gefitinib (10 mg/kg) was efficacious in inhibiting GBS penetration into the brain (Fig 1E). A high-degree of bacteremia is a primary determinant for GBS penetration into the brain (Ferrieri *et al*, 1980; Kim, 1987a, 1987b; Maruvada *et al*, 2011), but the magnitudes of bacteremia did not differ between the recipients of gefitinib and vehicle control. The decreased GBS penetration into the brain of the recipients of gefitinib is, therefore, not likely from having lower degrees of bacteremia compared to those of vehicle control. These *in vitro* and *in vivo* findings with gefitinib suggest that its target, EGFR, is likely to contribute to GBS invasion of the blood–brain barrier.

To support the contribution of EGFR to GBS invasion of the blood–brain barrier, we showed that EGFR activation occurred in response to GBS in a time- and inoculum-dependent manner in HBMEC (Figs 1F and EV3B). The role of EGFR in GBS invasion of the blood–brain barrier was next examined in HBMEC with EGFR knockout by CRISPR/Cas9. EGFR knockout HBMEC, as expected, exhibited no discernible EGFR expression (Fig 1G). We showed that eight CSF isolates of GBS belonging to prevalent hypervirulence clone sequence type ST-17 or less common ST-23 exhibited significantly decreased invasion in EGFR knockout HBMEC compared to control HBMEC (Fig 1G). We also showed that GBS transcytosis of HBMEC monolayer was inhibited in EGFR knockout cells compared to control HBMEC (Fig 1H). Gefitinib did not inhibit GBS invasion in EGFR knockout HBMEC, indicating the effect of gefitinib is specific to EGFR (Fig EV4A).

The role of EGFR in GBS penetration into the brain *in vivo* was examined using the tamoxifen-inducible, endothelial-specific EGFR knockout mice. Endothelial-specific conditional EGFR knockout was induced by a 5-day intraperitoneal administration of tamoxifen. 2 days later the mice were infected GBS. The bacterial counts recovered from the brains were significantly less in conditional EGFR knockout mice than in control Tek-RFP-Cre$^{ERT2}$ mice treated with tamoxifen (Fig 1I), while the bacterial counts from the blood did not differ between the two groups. These findings with EGFR knockout experiments demonstrate that meningitis isolates of GBS exploit EGFR for penetration of the blood–brain barrier *in vitro* and *in vivo*.

## Sphingosine-1 phosphate (S1P)-Sphingosine-1 phosphate receptor 2 (S1P$_2$) is upstream of EGFR in GBS penetration of the blood–brain barrier

It remains unclear how EGFR activation occurs in response to GBS and contributes to GBS penetration of the blood–brain barrier. We

determined whether inhibitors of specific host cell signaling molecules known to be involved in microbial invasion of the blood–brain barrier affected EGFR activation in response to GBS (Kim, 2008). We showed that EGFR activation in response to GBS was inhibited in HBMEC pretreated with JTE-013 (an antagonist of sphingosine 1-phosphate receptor 2, S1P$_2$) (Fig 2A). Sphingosine 1-phosphate (S1P) is known to function by binding to and signaling through its

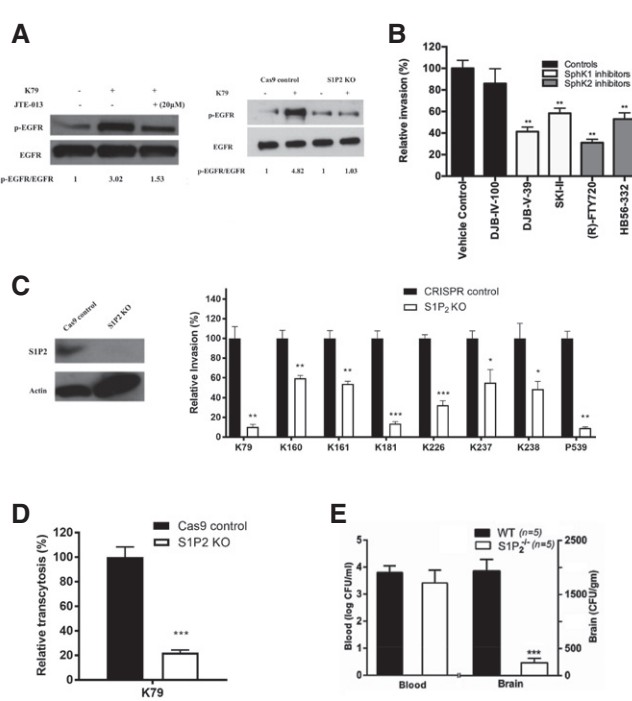

**Figure 2. Group B *Streptococcus* (GBS) exploits host S1P$_2$ for penetration of the blood–brain barrier.**

A  Tyrosine phosphorylation of EGFR in HBMEC pretreated with 20 μM S1P$_2$ antagonist (JTE-013) or vehicle control and infected with GBS strain K79 infection for 60 min. EGFR phosphorylation in control and S1P2 knockout HBMEC infected with K79.

B  Relative invasion frequency of strain K79 in HBMEC with or without SphK1 and SphK2 inhibitors. All the inhibitors were used at 10 μM. Data represent the means ± SEM from three independent experiments, with statistical analysis by Student's *t*-test. From left to right, *P* value of DJB-V-39 is 0.0047, *P* value of SKI is 0.0062, *P* value of FTY720 is 0.0022, and *P* value of HB56 is 0.0045.

C  S1P$_2$ protein expression in S1P$_2$ knockout HBMEC using CRISPR/Cas9 (left panel). Relative invasion frequency of 8 meningitis isolates of GBS in S1P$_2$ knockout HBMEC. Data represent the means ± SEM from three independent experiments, with statistical analysis by Student's *t*-test. From left to right, *P* value of K79 is 0.0018, *P* value of K160 is 0.0091, *P* value of K161 is 0.0054, *P* value of K181 is 0.00040, *P* value of K226 is 0.00032, *P* value of K237 is 0.040, *P* value of K238 is 0.041, and *P* value of P539 is 0.0025.

D  GBS strain K79 traversal of HBMEC monolayer was significantly decreased in S1P2 knockout HBMEC compare to control. Data represent the means ± SEM from three independent experiments, with statistical analysis by Student's *t*-test, **P = 0.00084.

E  Bacterial counts recovered from the blood and brain of wild-type (*n* = 5) and S1P$_2$$^{-/-}$ mice (*n* = 5) infected with strain K79 for 1h. Data represent the means ± SEM with statistical analysis by Student's *t*-test, ***P = 4.6E-05.

Source data are available online for this figure.

specific cell-surface receptors (S1P$_{1-5}$) (Maceyka *et al*, 2009; Blaho & Hla, 2014). There is no information on whether GBS exploits S1P for penetration of the blood–brain barrier.

We showed that S1P levels were significantly higher in HBMEC infected with GBS strain K79 compared to uninfected control, as measured by HPLC coupled tandem mass spectrometry after lipid extraction of HBMEC as previously described (Wang *et al*, 2016). The S1P content was normalized to lipid phosphate in the extracted samples and expressed as pmol/nmol lipid phosphate (mean $\pm$ SD of three samples), being at $2.75 \pm 0.37$ after 30 min at 37°C in HBMEC incubated with strain K79 vs. $1.19 \pm 0.45$ in uninfected cells ($P < 0.05$). S1P synthesis is catalyzed by sphingosine kinases 1 and 2 (SphK1 and SphK2) (Maceyka *et al*, 2009; Wang *et al*, 2016), and the role of S1P in GBS invasion of the blood–brain barrier was examined by determining the involvement of SphK1 and SphK2 using specific inhibitors (Wang *et al*, 2016). SphK1 inhibitors and SphK2 inhibitors significantly inhibited GBS invasion of HBMEC (Fig 2B), suggesting that S1P generation via SphK1 and SphK2 is likely to contribute to GBS invasion of the blood–brain barrier.

S1P is shown to function by interaction with its specific cell-surface receptors (S1P$_{1-5}$) (Maceyka *et al*, 2009; Blaho & Hla, 2014). We next determined the role of S1P receptors in GBS invasion of the blood–brain barrier using selective receptor antagonists (VPC23019 for S1P$_1$ and S1P$_3$, and JTE-013 for S1P$_2$) (Blaho & Hla, 2014; Wang *et al*, 2016). JTE-013 exhibited a dose-dependent inhibition of strain K79 invasion of HBMEC, while VPC23019 failed to exhibit such inhibition (Fig EV3C). These findings suggest that S1P interaction with S1P$_2$ is likely to play a role in GBS invasion of the blood–brain barrier. The contribution of S1P$_2$ to GBS invasion of the blood–brain barrier was examined in HBMEC with S1P$_2$ knockout by CRISPR/Cas9 (Fig 2C). Eight CSF isolates of GBS exhibited significantly decreased invasion in S1P$_2$ knockout HBMEC compared to control HBMEC (Fig 2C). We also showed that GBS penetration across HBMEC monolayer was inhibited in S1P2 knockout cells compared to control HBMEC (Fig 2D). Biological relevance of decreased GBS invasion in S1P$_2$ knockout HBMEC was examined in S1P$_2{}^{-/-}$ mice compared to wild-type C57BL/6j mice. GBS strain K79 penetration into the brain, as determined by the bacterial counts recovered from the brain, was significantly less in S1P$_2{}^{-/-}$ mice than in the wild-type animals without affecting the magnitudes of bacteremia (Fig 2E). EGFR activation in response to GBS was also inhibited in S1P2 knockout HBMEC (Fig 2A).

## EGFR is upstream of cPLA$_2\alpha$-CysLT1-ERM in GBS penetration of the blood–brain barrier

Our findings thus far indicate that GBS exploits host S1P-S1P$_2$-EGFR for penetration of the blood–brain barrier, but it remains unclear how EGFR contributes to GBS penetration. EGFR transactivation has been linked to cPLA$_2\alpha$ activation (Slomiany & Slomiany, 2009), and cPLA$_2\alpha$ activation is shown to be involved in GBS invasion of the blood–brain barrier (Maruvada *et al*, 2011). We hypothesize that EGFR exploits cPLA$_2\alpha$ for GBS invasion of the blood–brain barrier. This hypothesis is supported by our demonstration that cPLA$_2\alpha$ activation in response to GBS was inhibited in HBMEC pretreated with gefitinib and also in EGFR knockout HBMEC (Fig 3A and B). These findings demonstrate that EGFR is likely to function upstream of cPLA$_2\alpha$ in GBS invasion of the blood–brain barrier (EGFR-cPLA$_2\alpha$).

cPLA$_2\alpha$ release of arachidonic acid from the outer nuclear membrane is utilized for the biosynthesis of cysteinyl leukotrienes (CysLTs) (Peters-Golden & Henderson, 2007). We hypothesize that GBS might exploit CysLTs for penetration of the blood–brain barrier. CysLTs exhibit their biological actions via interaction with their G-protein coupled receptors, including CysLT1 and CysLT2 (Peters-Golden & Henderson, 2007). The roles of CysLTs in GBS invasion of the blood–brain barrier were determined initially by pharmacological inhibition, using the CysLT1 antagonist (montelukast) and the CysLT2 antagonist (BayCysLT2; Peters-Golden & Henderson, 2007). Pretreatment of HBMEC with montelukast significantly inhibited GBS invasion in a dose-dependent manner, while BayCysLT2 did not exhibit such a dose-dependent inhibition (Fig EV3D). CysLT1 knockdown HBMEC was used to demonstrate that the effect of

---

**Figure 3. GBS exploits host EGFR, S1P-S1P2, and cPLA2α-CysLT1-ERM for penetration of the blood–brain barrier.**

A    Serine phosphorylation of cPLA2α in response to GBS strain K79 in HBMEC pretreated with gefitinib or vehicle control and in EGFR knockout HBMEC.

B    Serine phosphorylation of cPLA2α in response to GBS strain K79 in control, EGFR knockout, and S1P2 knockout HBMEC.

C    Bacterial counts recovered from the blood and brain in wild-type mice (*n* = 5) and CysLT2$^{-/-}$ mice (*n* = 5) infected with strain K79 for 1h. Data represent the means $\pm$ SEM with statistical analysis by Student's *t*-test, *P* = 0.074.

D    CysLT1 protein expression in CysLT1 knockdown HBMEC using shRNA (left panel) and relative invasion frequency of 8 GBS isolates in CysLT1 knockdown and control HBMEC. Data represent the means $\pm$ SEM from three independent experiments, with statistical analysis by Student's *t*-test. From left to right, *P* value of K79 is 0.0039, *P* value of K160 is 0.0082, *P* value of K161 is 0.00091, *P* value of K181 is 0.0087, *P* value of K226 is 0.00020, *P* value of K237 is 0.0039, *P* value of K238 is 0.00066, and *P* value of P539 is 1.76E-05 (right panel).

E    GBS strain K79 traversal of HBMEC monolayer was significantly decreased in CysLT1 knockdown HBMEC compare to control. Data represent the means $\pm$ SEM from three independent experiments, with statistical analysis by Student's *t*-test, **P* = 0.0021.

F    Bacterial counts recovered from the blood and brain in wild-type mice (*n* = 7) and CysLT1$^{-/-}$ mice (*n* = 7) infected with strain K79 for 1 h. Data represent $\pm$ SEM with statistical analysis by Student's *t*-test, **P* = 0.021.

G    Ezrin protein expression in ezrin knockdown HBMEC using shRNA (left panel) and relative invasion frequency of GBS strain K79 in ezrin knockdown and control HBMEC (right panel). Data represent the means $\pm$ SEM from three independent experiments, with statistical analysis by Student's *t*-test, **P* = 0.0044.

H, I    Ezrin phosphorylation in HBMEC pretreated with CysLT1 antagonist (montelukast) and infected with GBS strain K79 for 60 min (left panel) and ezrin phosphorylation in the homogenates of brain capillaries of CysLT1$^{-/-}$ and control mice infected with GBS strain K79 for 1h (right panel).

J    Intracellular K79 co-localization with EGFR and ezrin in control and S1P2 knockout HBMEC (as shown by arrows, cyan). Percentage of co-localization was calculated by counting numbers of all GBS and co-localized intracellular GBS from at least three representative fields, and expressing as numbers of co-localized GBS/all GBS × 100. Data represent the means $\pm$ SEM of at least 60 GBS from three fields, with statistical analysis by Student's *t*-test. In left panel, **P* = 0.034. In right panel, **P* = 0.0048.

Source data are available online for this figure.

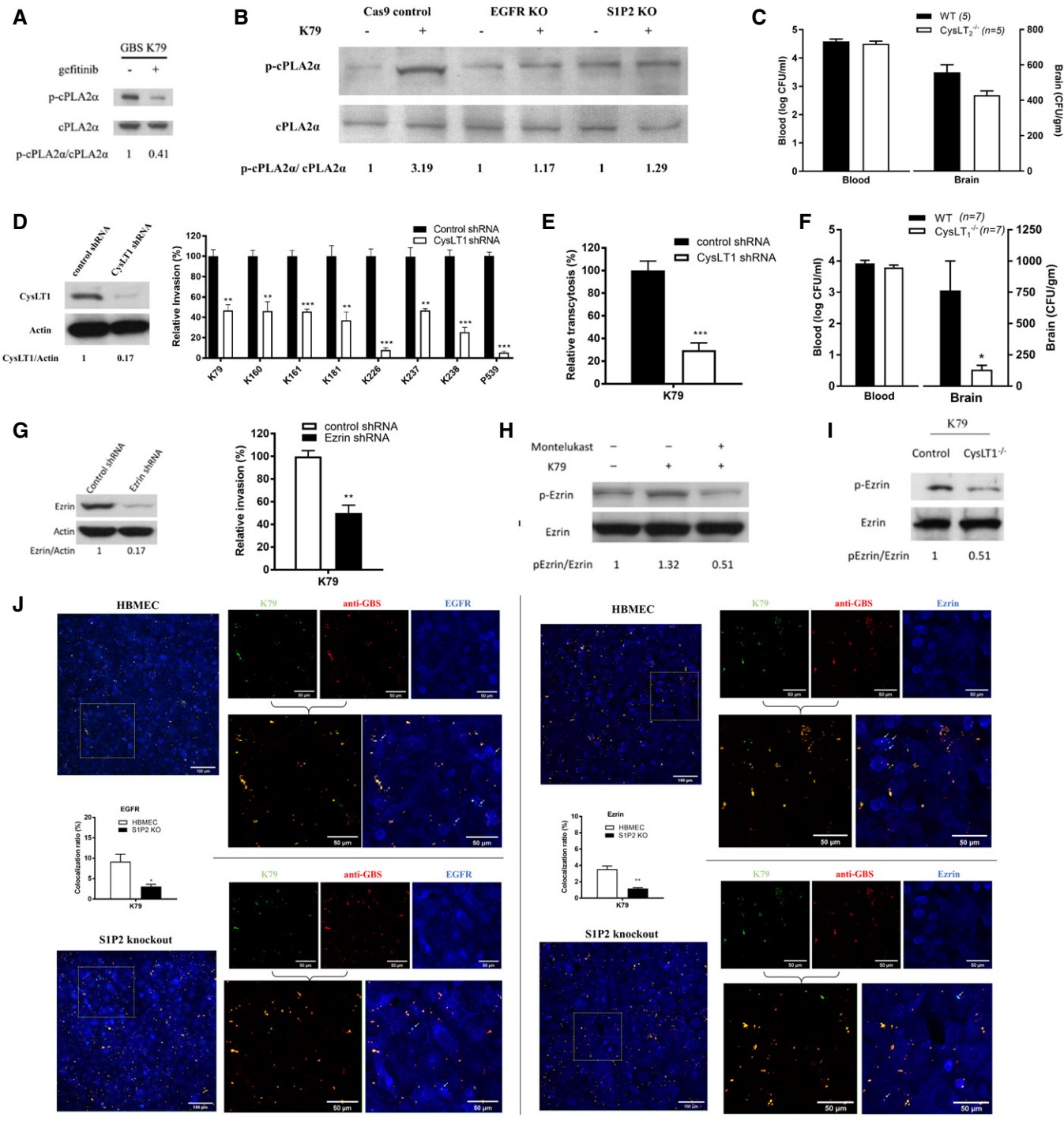

**Figure 3.**

montelukast was specific to CysLT1, as shown by the demonstration that montelukast did not inhibit GBS invasion in CysLT1 knock-down cells (Fig EV4A). The selective activity of BayCysLT2 on CysLT2 is dose-dependent, involving 0.1 μM or less, and BayCysLT2 doses greater than 1 μM are shown to affect CysLT2 and CysLT1. BayCysLT2 inhibition of GBS invasion of HBMEC at 10 μM is, therefore, likely due to its action on CysLT1. This concept is supported by the demonstration that GBS penetration into the brain

was not affected in CysLT2$^{-/-}$ compared to wild-type animals, as the bacterial counts recovered from the brains did not differ signifi-cantly between CysLT2$^{-/-}$ and wild-type animals (Fig 3C). It is important to note that montelukast at 50 μM and BayCysLT2 at 10 μM did not affect the growth of GBS strain K79 and also did not affect the integrity of HBMEC monolayer, as assessed by live/dead staining (Molecular Probes). These findings demonstrate that GBS is likely to exploit CysLT1 for invasion of the blood–brain barrier.

The contribution of CysLT1 to GBS invasion of HBMEC monolayer was demonstrated by significantly less invasion of eight meningitis isolate of GBS in CysLT1 knockdown HBMEC compared to control HBMEC (Fig 3D). We also showed that GBS penetration across HBMEC monolayer was inhibited in CysLT1 knockdown HBMEC compared to control HBMEC (Fig 3E). We next examined the effect of pharmacological inhibition and gene deletion of CysLT1 in GBS penetration into the brain in experimental animal model of hematogenous meningitis. Administration of montelukast (5 mg/kg, a dose which exhibits CysLT1 antagonist activity in mice) (Zhu *et al*, 2010) intraperitoneally 1 h before intravenous injection of GBS significantly inhibited GBS penetration of the brain of BALB/c mice, as shown by significantly decreased bacterial counts recovered from the brains of the recipients of montelukast compared to those of vehicle control (Fig EV3E). This finding is consistent with that of a previous report demonstrating that montelukast inhibited GBS penetration into the brain (Syu *et al*, 2019). The contribution of CysLT1 to GBS penetration into the brain was verified in CysLT1 knockout animals, as the bacterial counts recovered from the brain were significantly less in CysLT1$^{-/-}$ mice than in the wild-type animals (Fig 3F). In contrast, the levels of bacteremia did not differ between the recipients of montelukast and vehicle control, and also between CysLT1$^{-/-}$ and wild-type animals, as shown by similar numbers of bacterial counts recovered from the blood (Figs 3F and Fig EV3E), indicating that decreased GBS penetration into the brains of animals receiving montelukast and CysLT1$^{-/-}$ mice did not stem from lower degrees of bacteremia compared to control animals. These findings demonstrate that GBS exploits host CysLT1 for penetration of the blood–brain barrier *in vitro* and *in vivo*.

Our findings demonstrate that EGFR is upstream of cPLA$_2$α-CysLT1 in GBS invasion of the blood–brain barrier (EGFR-cPLA$_2$α-CysLT1). It remains, however, unclear how host CysLT1 contributes to GBS invasion of the blood–brain barrier. CysLT1 represents G-protein coupled receptor and regulates diverse cellular responses in endothelial cells including stress fiber formation (Duah *et al*, 2013). ERM are key proteins linking plasma membrane to actin filaments (Arpin *et al*, 2011; Kawaguchi *et al*, 2017), and we hypothesize that CysLT1 exploits ERM in host cell actin cytoskeleton rearrangements, a prerequisite for GBS invasion of HBMEC monolayer (Nizet *et al*, 1997; Kim, 2008). This hypothesis is supported by our demonstration that ezrin contributes to GBS invasion of blood–brain barrier, as evidenced by significant reduction of GBS invasion in the ezrin shRNA knockdown HBMEC (Fig 3G). The concept of CysLT1 exploitation of ezrin in GBS invasion of HBMEC was also supported by our demonstration that ezrin activation occurring in response to GBS was inhibited in HBMEC pretreated with montelukast (Fig 3H). In addition, ezrin activation occurring in homogenates of the brain capillaries derived from infected wild-type BALB/c mice was inhibited in homogenates of the brain capillaries of infected CysLT1$^{-/-}$ mice (Fig 3I). These *in vitro* and *in vivo* findings support that ezrin is likely to be involved in CysLT1-mediated GBS invasion of the blood–brain barrier (CysLT1-ezrin).

We showed that EGFR is upstream of cPLA$_2$α-CysLT1 in GBS penetration of the blood–brain barrier. Since S1P-S1P$_2$ is shown to be upstream of EGFR, we examined whether cPLA$_2$α activation in response to GBS (which is downstream of EGFR, Fig 3A) can be inhibited in S1P$_2$ knockout HBMEC compared to control HBMEC.

cPLA$_2$α activation in response to GBS was, as expected, inhibited in S1P$_2$ knockout HBMEC (Fig 3B). Taken together, these *in vitro* and *in vivo* findings demonstrate for the first time that GBS exploits host S1P-S1P$_2$, EGFR, and cPLA$_2$α-CysLT1-ezrin for penetration of the blood–brain barrier and that S1P-S1P$_2$ and cPLA$_2$α-CysLT1-ezrin represent upstream and downstream molecules of EGFR, respectively, in GBS penetration of the blood–brain barrier (synopsis figure). This concept is also supported by the demonstration of co-localizations of internalized GBS strain K79 with EGFR and ezrin in control HBMEC, while such co-localizations were significantly less in S1P$_2$ knockout HBMEC compared to control HBMEC (Fig 3J).

### Targeting host factors identified from investigating GBS invasion of the blood–brain barrier improves the outcome of animals with experimental GBS meningitis

Antimicrobial therapy alone has limited efficacy in improving the outcome of animals with experimental hematogenous GBS meningitis (Kim, 1987b). At present, no strategy exists for development of an improved therapy for GBS meningitis. We, therefore, examined whether blockade of targets derived from investigating GBS invasion of the blood–brain barrier might represent a beneficial adjunct to antimicrobial therapy in improving the outcome of animals with GBS meningitis. This issue was examined using pharmacological antagonist of CysLT1 in experimental settings mimicking clinical scenario, e.g., animals with established GBS meningitis were treated with an antibiotic (ceftriaxone), CysLT1 antagonist (montelukast), or combination of ceftriaxone and montelukast. CysLT1 antagonists have been studied extensively in clinical trial (Kemp, 2005). Animals received intravenously GBS and 24 h later were randomly divided to receive ceftriaxone and montelukast alone or in combination daily for 7 days (randomization was done at the time of GBS inoculation to minimize any selection bias) and assessed for survival, neuronal injury and apoptosis, astrocytes and microglial activation, and memory function. All animals with GBS meningitis receiving montelukast alone died within 2 days of treatment. Survival was significantly greater ($P = 0.015$) in animals with GBS meningitis receiving ceftriaxone and montelukast compared to animals receiving ceftriaxone alone (Fig 4A), suggesting that counteracting CysLT1 as an adjunctive therapy was beneficial in improving survival of animals receiving ceftriaxone.

Neuronal injury occurs commonly in survivors of GBS meningitis (Kim, 2010; Romain *et al*, 2018; O'Sullivan *et al*, 2019). Animals with GBS meningitis receiving ceftriaxone and montelukast exhibited significantly less neuronal injury as assessed by Nissl and TUNEL stains (Fig 4B and C), less activation of microglia and astrocytes (Fig 4D), and less memory impairment as assessed by Y-maze (Fig 4E).

We next examined whether inhibition of other host factors in the network was beneficial in improving the survival of animals with GBS meningitis and particularly whether inhibition of host factors together was more efficacious in survival than inhibition of individual factors by the method described above. Pharmacological inhibition of EGFR in combination with antibiotic therapy significantly improved the survival compared to antibiotic therapy alone. However, survival did not differ between animals receiving ceftriaxone plus inhibitors of both CysLT1 and EGFR vs single inhibitors (Fig EV4B). These findings are consistent with those of our *in vitro*

experiments, where gefitinib did not further decrease GBS invasion in CysLT1 knockdown HBMEC and montelukast did not exhibit additional decrease in GBS invasion in EGFR knockout HBMEC (Fig EV4A). Taken together, these findings demonstrate that GBS penetration of the blood–brain barrier exploits S1P2-EGFR-CysLT1

and suggest that contributions of individual host factors to GBS meningitis are not additive.

Taken together, GBS penetration of the blood–brain barrier exploits a defined network comprised of specific host factors (S1P$_2$, EGFR, and CysLT1). We showed that counteracting such network,

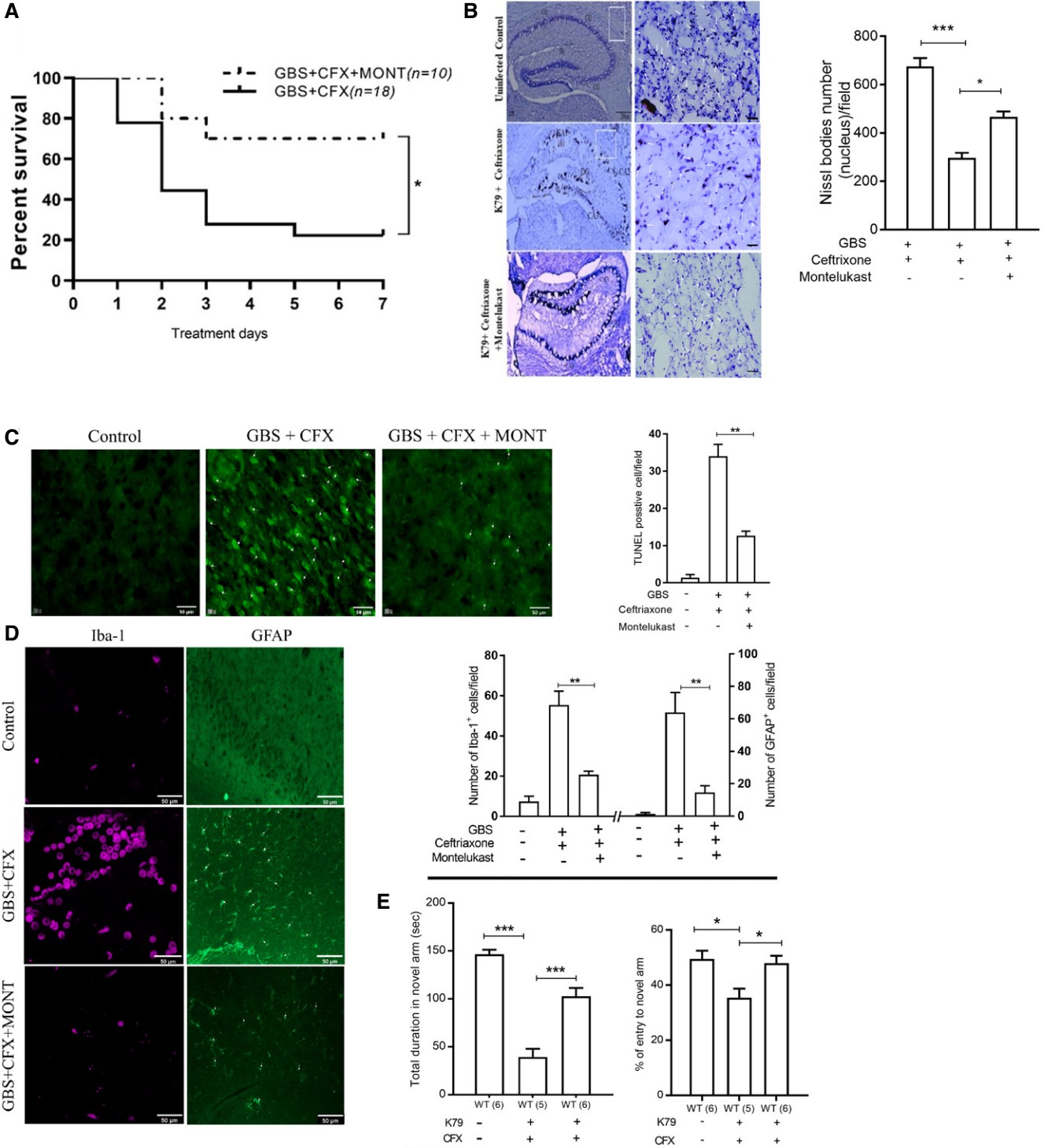

Figure 4.

**Figure 4. Use of montelukast as an adjunct to an antibiotic (ceftriaxone) in therapy of experimental GBS meningitis.**

A Survival of wild-type mice receiving drug administration for 7 days after infection with GBS strain K79. Data are presented as a Kaplan–Maier plot with a log-rank test used to compare percentage of survival between the groups, CFX group $n = 18$, and CFX + MONT group $n = 10$ *$P = 0.015$. (CFX: ceftriaxone; MONT: montelukast).

B Nissl staining of the hippocampus CA1, CA2, and CA3 regions (low magnification) and cortex region (higher magnification) (left) and histograms showing the number of Nissl-stained bodies (right) Scale bar, 50 μm. Data represent the means ± SEM from three independent experiments, with statistical analysis by one-way ANOVA, $P$ value between control group and Ceftriaxone group is 0.00021, $P$ value between Ceftriaxone group and Ceftriaxone + Montelukast group is 0.013.

C, D Representative immunofluorescence images (left) and quantification (right) of TUNEL bodies, astrocyte (GFAP), and microglia (Iba1) in cortex of wild-type mice after 7 days of therapy. Scale bar, 50 μm. Data represent the means ± SEM from three independent experiments, with statistical analysis by one-way ANOVA, $P$ value of TUNEL bodies between CFX group and MONT + CFX group is 0.0012, $P$ value of astrocyte between CFX group and MONT + CFX group is 0.0068, and $P$ value of microglia cells between CFX group and MONT + CFX group is 0.0029.

E Y-maze test for spatial learning and memory of different group of wild-type mice. The total duration of time spent at novel arm (left) and the percentage of entries to novel arm of Y-maze (right). Data represent the means ± SEM was statistically analyzed by one-way ANOVA, $P$ value of total duration of time spent at novel arm between control group ($n = 6$) and CFX group ($n = 5$) is 0.00012, $P$ value between CFX group ($n = 5$) and MONT + CFX group ($n = 6$) is 0.00017; $P$ value of total duration of % of entry to novel arm between control group ($n = 6$) and CFX group ($n = 5$) is 0.0102, and $P$ value between CFX group ($n = 5$) and MONT + CFX group ($n = 6$) is 0.0216.

Source data are available online for this figure.

as exemplified with pharmacological inhibition of CysLT1, was a beneficial adjunct to antibiotic therapy of GBS meningitis, as shown by improved survival, decreased neuronal injury and neuroinflammation, resulting in less functional impairment and neurological sequelae.

## Discussion

GBS remains the leading cause of neonatal bacterial meningitis and GBS meningitis continues to be an important cause of mortality and morbidity (Kim, 2010; Romain *et al*, 2018; O'Sullivan *et al*, 2019). A major contributing factor to such mortality and morbidity is our incomplete understanding of the pathogenesis of GBS meningitis. Meningitis isolates of GBS exhibit the ability to invade HBMEC monolayer and penetrate into the brain (Kim, 2008; Maruvada *et al*, 2011). Group B *Streptococcus* strains have been shown to invade HBMEC monolayer without affecting the integrity of HBMEC monolayer (Nizet *et al*, 1997; Maruvada *et al*, 2011), and GBS penetration into the brain occurs without affecting the blood–brain barrier permeability and influx of immune cells (Kim *et al*, 1997; Doran *et al*, 2005; Tazi *et al*, 2010). This concept was recapitulated in this study, where GBS penetration occurred initially in the meningeal and cortex capillaries, and subsequent invasion into the brain without extravasation of intravascular small molecule tracer, which was shown under a stringent condition for assessing the blood–brain barrier permeability using a small molecule of 443Da as well as without affecting structural integrity of the blood–brain barrier. These findings support that GBS strains exploit a transcellular penetration of the blood–brain barrier and elucidation of GBS invasion of the blood–brain barrier is likely to enhance the pathogenesis of GBS meningitis. GBS serotype III strains are common in causing meningitis, and ST-17 is predominant (70–80%) in meningitis isolates of GBS (Tazi *et al*, 2010), but it remains incompletely understood how GBS strains including those belonging to ST-17 penetrate the blood–brain barrier.

Meningitis-causing pathogens exploit host cell signaling molecules to promote their penetration of the blood–brain barrier (Kim, 2008), but the underlying mechanisms vary depending upon the pathogens. Our findings reported here demonstrate that type III GBS strains belonging to ST-17 and less common ST-23 isolated from

neonates with meningitis exploit a network of specific host cell signaling molecules (S1P$_2$, EGFR, and CysLT1) for penetration of the blood–brain barrier. The proposed network comprised of S1P$_2$, EGFR, and CysLT1 has not been previously recognized for their contribution to GBS meningitis and this is the first report to demonstrate the contribution of such network to GBS penetration of the blood–brain barrier *in vitro* and *in vivo*.

A high-degree of bacteremia is shown to be a primary determinant for GBS penetration of the blood–brain barrier *in vivo*, and the mechanisms involved with a high-degree of bacteremia for GBS invasion of the BBB remain incompletely understood. Our data showed that EGFR activation in response to GBS was inoculum-dependent, suggesting that one mechanism requiring a high-degree of bacteremia is related to dose-dependent activation of host cell signaling molecules involved in GBS invasion (Fig EV3B). It is important to document that decreased GBS penetration into the brain occurring with inhibition and/or blockade of the above-mentioned host factors is not due to a lower magnitude of bacteremia (Kim, 1987b; Maruvada *et al*, 2011). Pharmacological inhibition and gene deletion of S1P$_2$, EGFR, and CysLT1 did not affect the magnitudes of GBS bacteremia, but resulted in decreased GBS penetration into the brain, indicating that the contributions of S1P$_2$, EGFR, and CysLT1 to GBS penetration into the brain are not due to lower magnitudes of bacteremia.

Pharmacological inhibition can point to specific targets of interest such as gefitinib for EGFR, JTE-013 for S1P$_2$, and montelukast for CysLT1, but not without any concern about off-target effects. This off-target issue was addressed by targeting specific genes such as knockout and knockdown approaches in HBMEC as well as using gene knockout animals, e.g., CRIPSR/Cas9 knockout of EGFR and S1P$_2$, shRNA knockdown of CysLT1, and knockout animals for S1P$_2$, EGFR, and CysLT1. The specificity of inhibitors for their targets was also demonstrated by no inhibition of GBS invasion in respective target knockout/knockdown HBMEC. For example, gefitinib and montelukast inhibited GBS invasion in control HBMEC, but not in EGFR knockout and CysLT1 knockdown HBMEC, respectively. These findings indicate that S1P$_2$, EGFR, and CysLT1 are indeed involved in GBS penetration of the blood–brain barrier *in vitro* and *in vivo*.

Of a particular novel finding is our demonstration that the contributions of S1P$_2$, EGFR, and cPLA$_2$α-CysLT1-ezrin to GBS penetration

of the blood–brain barrier were inter-related. This concept is shown by the demonstration that (i) pharmacological inhibition and knockout of $S1P_2$ and EGFR inhibited $cPLA_2\alpha$ activation in response to GBS and (ii) pharmacological inhibition and knockout of $S1P_2$ resulted in inhibition of EGFR activation in HBMEC. These findings demonstrate that S1P2 is upstream of EGFR, while $cPLA_2\alpha$-CysLT1 is downstream of EGFR in GBS penetration of the blood–brain barrier. In addition, we showed that ezrin is downstream of CysLT1, as shown by inhibition of ezrin activation in response to GBS in HBMEC pretreated with the CysLT1 antagonist and in brain capillaries derived from CysLT1$^{-/-}$ mice, demonstrating for the first time that CysLT1 exploits ezrin in GBS invasion of the blood–brain barrier. In addition, our determinations of host factors contributing to GBS penetration of the blood–brain barrier by themselves ($S1P_2$, EGFR, and CysLT1) and by elucidating up and downstream connections through our defined network ($S1P_2$-EGFR-CysLT1) support the contribution of such network to GBS penetration of the blood–brain barrier.

We have shown that EGFR-CysLT1 contributes to *E. coli* invasion of the blood–brain barrier (Zhu *et al*, 2020), but the mechanisms involved with *E. coli* differ from those with GBS. For example, host cell signaling pathway exploited by *E. coli* invasion involves both the EGFR-CysLT1-dependent and independent pathways, as shown by the demonstration that decreased *E. coli* invasion in EGFR and CysLT1 knockout/knockdown HBMEC can be further reduced by montelukast and gefitinib, respectively. In contrast, GBS invasion was dependent upon the EGFR-CysLT1 pathway, as shown by no additional inhibition of GBS invasion by gefitinib and montelukast in CysLT1 and EGFR knockdown/knockout HBMEC, respectively (Fig EV4A and C). Additional studies are needed to elucidate the molecular basis of $S1P_2$, EGFR, and CysLT1 for their contribution to GBS penetration of the blood–brain barrier.

More importantly, our network-based targeted approach was beneficial in development of an adjunct therapy to an antibiotic therapy in improving the outcome of animals with established GBS meningitis. This concept was shown by improved survival, less neuronal injury, less activation of microglia and astrocytes, and less memory impairment in animals with GBS meningitis who received the combination therapy of an antibiotic and a CysLT1 antagonist compared to those receiving antibiotic therapy alone. This issue is particularly relevant to GBS meningitis because GBS strains are uniformly susceptible to penicillins and poor outcome of GBS meningitis is not due to emergence of non-susceptible GBS strains, necessitating development of an adjunct therapy for GBS meningitis. At present, no strategy exists to develop a target(s) for adjunct therapy. Our findings indicate that development of an adjunct therapy can be feasible by counteracting targets identified from investigating GBS invasion of the blood–brain barrier (e.g., SphK, EGFR, CysLT1). SphK inhibitors, EGFR inhibitors, and CysLT1 antagonists have been studied in clinical trials (Mendelsohn & Baselga, 2003; Kemp, 2005; Britten *et al*, 2017; Thomas *et al*, 2017), and it remains to be determined whether inhibition of such targets exhibits therapeutic benefits similar to those of CysLT1 in improving the outcome of experimental GBS meningitis. It is important to acknowledge that administration of two inhibitors together did not provide additional improved survival compared to individual inhibitors. Studies are needed to elucidate how targets derived from investigating GBS invasion of the blood–

brain barrier can function as adjunct therapeutic targets for GBS meningitis. An adjunct therapy for GBS meningitis is likely to be used once for a brief period of time (e.g., our regimen for 7 days), minimizing any associated adverse effects of potential drugs, and clinical trials will determine whether targets shown in this report is beneficial as an adjunct therapy.

Taken together, this is the first demonstration that meningitis isolates of GBS exploit $S1P_2$ for EGFR activation and exploit EGFR for $cPLA_2\alpha$-CysLT1 activation in penetration of the blood–brain barrier *in vivo* and *in vitro*, indicating that $S1P_2$-EGFR-$cPLA_2\alpha$-CysLT1 represents a novel host cell signaling network for investigating the pathogenesis and therapy of GBS meningitis.

# Materials and Methods

### GBS strains

Eight GBS type III strains, belonging to hypervirulence clone sequence type 17 (ST-17), strains K79, K160, K161, K181, K226, K237, K238, accounting for 70–80% of GBS meningitis, and less common ST-23, strain P539, were derived from neonates with meningitis, as previously described (Kim, 1987a, 1987b; Maruvada *et al*, 2011).

### Reagents

Arachidonyltrifluoromethyl ketone (AACOCF3; $cPLA_2$ inhibitor) was purchased from Biomol Laboratories (Plymouth Meeting, PA). Montelukast (cysteinyl leukotriene type 1 receptor antagonist), BayCysLT2 (cysteinyl leukotriene type 1 receptor antagonist), JTE-013 ($S1P_2$ antagonist), VPC23019 ($S1P_1$ and $S1P_3$ antagonist), and gefitinib (EGFR tyrosine kinase inhibitor) were purchased from Cayman Chemical Company (Ann Arbor, MI). $cPLA_2$ (1:1,000) and phospho-$cPLA_2\alpha$ (1:1,000) antibodies were purchased from Cell Signaling Technologies (Danvers, MA). Cysteinyl leukotriene type 1 receptor (CysLT1) (1:1,000), CD31 (1:200 for tissue staining), and EGFR (1:1,000 for Western blot, and 1:200 for IHC) antibodies were purchased from Santa Cruz Biotechnology (Santa Cruz, CA). DAPI, anti-phospho-ezrin antibody (1:1,000), anti-claudin-5 (1:200 for tissue staining), anti-mouse IgG Alexa Fluor 488-conjugated secondary antibody (1:1,000), and EZ-Link™ Sulfo-NHS-Biotin were from Thermo Fisher Scientific (Waltham, MA). Anti-phosphotyrosine antibody clone 4G10 (used at 1:1,000) and anti-ezrin (1:2,000) were purchased from Millipore Sigma (St Louis, MO). The anti-$S1P_2$ (1: 500) antibody was purchased from Proteintech (Rosemont, IL). Iba-1 antibody (1:200 for tissue staining) was purchased from Abcam (Cambridge, MA). The glial fibrillary acidic protein GFAP antibody (1:200 for tissue staining) was purchased from Dako (Denmark). The Texas Red® Streptavidin (1:1,000) was purchased from Vector Laboratories (Burlingame, CA).

### RNA-seq and gene expression analysis

The HBMEC transcriptional responses to GBS were determined as previously described (Káňová *et al*, 2019). Briefly, the HBMEC ($1.2 \times 10^6$ cells/dish) were seeded in 100 mm dishes and cultured for 4 days till confluence in RPMI medium supplemented with 10%

FBS, 10% Nu-serum, 1% MEM vitamin, 1% MEM non-essential amino acids, 1 mM sodium pyruvate, and 100 U/ml penicillin–streptomycin. The HBMEC was incubated for 1 h in experimental medium composed of Medium 199 and Ham's F12 (1:1) supplemented with 5% FBS before infection assay. GBS strain K79, $3 \times 10^8$ cells/dish at a multiplicity of infection (MOI) of 100 was loaded into 100-mm dishes and incubated for 2 h and 6 h RNA was purified from HBMEC using Quick-RNA Microprep (ZYMO Research, R1050).

Total RNA was purified from HBMEC for RNA-seq analysis as described previously (Liu *et al*, 2015; Watkins *et al*, 2018). All RNA-seq libraries (non-strand-specific, paired end) were prepared using TruSeq RNA Sample Prep kit (Illumina). The total RNA samples isolated from HBMEC infected with GBS were subject to poly(A) enrichment as part of the TruSeq protocol. One hundred nucleotides of sequence were determined from each end of each cDNA fragment using the HiSeq platform (Illumina). Sequencing reads were annotated and aligned to the Ensembl GRCh38 of the human reference genome (Cunningham *et al*, 2015) using TopHat2 (Kim *et al*, 2013). The alignment files from TopHat2 were used to generate read counts for each gene, and a statistical analysis of differential gene expression was performed using the edgeR package from Bioconductor (Robinson *et al*, 2010). Cut-offs include a *P* value of 0.05 or less and an absolute $\log_2$ fold change of 1. The values are log2 fold change of infected vs uninfected HBMEC. The Ingenuity Pathway Analysis (IPA) software (Ingenuity Systems; http://www.ingenuity.com) was used for upstream regulator analysis on the sets of host genes that were differentially expressed (*P* < 0.05) between the infection groups and the uninfected control group (Liu *et al*, 2015; Watkins *et al*, 2018). This software could assess the overlap between RNA-seq derived gene lists and an extensively curated database of target genes for each of several hundred known regulatory proteins. It then uses the statistical significance of the overlap and the direction of differential gene expression to make predictions about activation or repression of these regulatory proteins. These proteins were then performed a protein–protein interaction enrichment analysis using Metascape (http://metascape.org). It utilizes physical protein–protein interactions captured in BioGrid as the main data source and also it integrates more recent human interactome datasets including InWeb_IM and OmniPath to provide additional interactome coverage (Zhou *et al*, 2019). The visualized protein interaction presented the connection between these host factors and revealed the potential signal transduction networks.

For HBMEC, short tandem repeat (STR) profiling was used to authenticate HBMEC lines used in this manuscript using FTA Sample Collection Kit for human cell authentication service, ATCC 135-XV, and there was no mycoplasma contamination.

## Mice

GBS penetration into the brain in the presence and absence of pharmacological inhibitors was examined in wild-type BALB/c and C57BL/6 mice, both male and female, between 4 and 5 weeks old. We also used specific knockout mice along with their strain-matched wild-type animals for delineating specific host factors in GBS penetration into the brain; $CysLT1^{-/-}$, $CysLT2^{-/-}$ mice and BALB/c, and $S1P_2^{-/-}$ and C57BL/6 mice (Kono *et al*, 2007;

Maekawa *et al*, 2008). All procedures and handling techniques were approved by The Johns Hopkins Animal Care and Use Committee.

## Construction of EGFR conditional knockout mice

The EGFR knockout mice are embryonically lethal, and the tamoxifen-inducible, endothelial-specific EGFR knockout mice were generated by crossbreeding floxed EGFR mice (Lee & Threadgill, 2009) with Tek-RFP-Cre^ERT2 mice in the background of C57BL/c as described previously (Chen *et al*, 2015; Zhao *et al*, 2018). EGFR^fl/fl mice were crossed with Tek-RFP-Cre^ERT2 mice to generate EGFR^fl/fl /Tek-RFP-Cre^ERT2mice. Genotyping was performed by PCR of tail DNA with the Cre primers (Cre forward: 5'- CTA AAC ATG CTT CAT CGT CGG TC −3'; Cre reverse: 5'- TCT GAC CAG AGT CAT CCT TAG CG −3') and Lox3 primers (Lox3 forward: 5'- CTT TGG AGA ACC TGC AGA TC −3'; Lox3 reverse: 5'- CTG CTA CTG GCT CAA GTT TC −3'). Cre^ERT2 consists of Cre conjugated with a mutated estrogen receptor (ERT2) that binds poorly to endogenous estrogen but with high affinity to the estrogen derivative tamoxifen (Thomas *et al*, 2017). Endothelial-specific knockout of EGFR was induced by a 5-day intraperitoneal administration of tamoxifen (1 mg/kg). 2 days later, the mice were infected with GBS, and the bacteria counts from the blood and brain were determined as described above. Tek-RFP-Cre^ERT2 littermates treated with tamoxifen were served as controls.

## GBS invasion assays in HBMEC

HBMEC were isolated, characterized, and used for GBS invasion assays as described previously (Nizet *et al*, 1997; Stins *et al*, 1997, 2001; Maruvada *et al*, 2011). Briefly, GBS strains grown overnight in Todd–Hewitt broth (Difco Laboratories, Detroit, MI) were resuspended in experimental medium [M199-HamF12 (1:1) containing 5% heat-inactivated fetal bovine serum, 2 mM glutamine, and 1 mM pyruvate] and added in a MOI of 100 to HBMEC. After 2 h of incubation at 37°C, HBMEC were washed with RPMI 1640 and incubated with experimental media containing penicillin (10 μg/ml) and gentamicin (100 μg/ml) for 1 h to kill extracellular bacteria. The cells were washed again with PBS, lysed in 0.025% Triton X-100, and the released intracellular bacteria were enumerated by plating on sheep blood agar plates. The invasion results were calculated as a percent of the initial inoculum and expressed as percent relative invasion compared to percent invasion of GBS in the presence of vehicle control (DMSO). Each set was run in triplicates.

## Assays for GBS penetration across HBMEC monolayer

HBMEC were cultured on Transwell polycarbonate tissue culture inserts with a pore diameter of 8 μm (Corning Costar) for 5 days. On the morning of the assay, HBMEC monolayer was washed with experimental medium and GBS (MOI = 10) were added to the upper chamber (Nizet *et al*, 1997; Stins *et al*, 2001). After 1.5 h of incubation at 37°C, samples were taken from the lower chamber and plated for determinations of CFUs. The integrity of the HBMEC monolayer was assessed by measurements of the transendothelial electrical resistance (TEER) before and after assays as well as live/

dead staining (Molecular Probes), as previously described (Stins et al, 2001).

## Immunoblotting and immunoprecipitation

The lysates of HBMEC and homogenates of the mouse brain capillaries with and without GBS infection were prepared for Western blotting and immunoprecipitation as described previously (Stins et al, 1997; Das et al, 2001; Zhu et al, 2010; Maruvada et al, 2011; Wang et al, 2016).

## Measurement of sphingosine 1-phosphate (S1P) in HBMEC by HPLC coupled tandem mass spectrometry

HBMECs were grown to confluence in 100-mm tissue culture dish, serum-starved overnight, and infected with GBS strains K79 at a MOI of 100 for 30 min at 37°C. Sphingolipids of the HBMECs were extracted and quantified by HPLC coupled tandem mass spectrometry as previously described (Wang et al, 2016).

## Construction of EGFR and S1P$_2$ CRISPR knockout HBMECs

Cas9 gene was incorporated into the HBMECs by lentivirus produced by lentiCas9-Blast plasmid (Addgene #52962) in 293T (Wang et al, 2016; Joung et al, 2017). The EGFR target DNA sequences and S1P$_2$ target DNA sequences were cloned into LentiGuide-puro plasmid (Addgene #52963) using primers EGFR-CRISPR-3A-F CACCGTGCGCTCTGCCCGGCGAGTC, EGFR-CRISPR-3A-R AAACGACTCGCCGGGCAGAGCGCAC, EGFR-CRISPR-4B-F CACCGCCGGCTCTCCCGATCAATAC, EGFR-CRISPR-4B-R AAACGTATTGATCGGGAGAGCCGGC: primers S1P2-CRISPR-1F CACCGCAGCTCTCCAGTGGGAGGA T, S1P2-CRISPR-1R AAACATCCTCCCACTGGAGAGCTGC, S1P2-CRISPR-2F CACCGCGCCTGTAATCCCAGCAATT, S1P2-CRISPR-2F AAACAATTGCTGGGATTACA GGCGC. The 293T cells were transfected with 10 μg of the transfer plasmid LentiGuide-puro-EGFR or S1P$_2$, 5 μg pVSV-G, 7.5 μg psPAX2, 100 μl of Plus Reagent (Life Technologies), and 50 μl of Lipofectamine 2000 (Life Technologies) to produce the CRISPR lentivirus targeting EGFR or S1P$_2$. HBMEC-Cas9 cells were infected by CRISPR lentivirus targeting EGFR or S1P$_2$ and selected by HBMEC complete media supplemented with 1 μg/ml puromycin for CRISPR knockout cells. The single cells of EGFR or S1P$_2$ CRISPR knockout HBMECs were sorted to 96-well tissue culture plate by flow cytometry and grown to confluence in HBMEC complete media supplemented with 1 μg/ml puromycin. The expression of EGFR and S1P$_2$ was verified by Western blotting with anti-EGFR and S1P$_2$ antibodies, respectively.

## Construction of CysLT1 and ezrin shRNA knockdown HBMECs

The shRNA lentiviral plasmids targeting CysLT1 or ezrin and green fluorescent protein (shRNA control) were purchased from TRC shRNA libraries at High Throughput Biology Genomic Resources of Johns Hopkins University (http://hitcores.bs.jhmi.edu/). The 293T cells (ATCC CRL-3216) were cultured at 37 °C and 5% CO$_2$ in Modified Eagle's Medium (DMEM) (ATCC # 302002) complemented with heat inactivated 10% fetal bovine serum (FBS) (ATCC # 302020), 2 mM L-glutamine (ATCC # 302214), and 1%

penicillin/streptomycin. During transfection, $1.5 \times 10^6$ 293T cells were cultured in 6-well tissue culture plates overnight in 5 ml Opti-MEM (Life Technology # 31985070) and transfected with 3 μg lentiviral plasmids containing the target DNA sequence, 2 μg packaging plasmid psPAX2 (Addgene # 12260), 1 μg envelop plasmid pCMV-VSV-G (Addgene # 8454), and 6 μl Fugene Transfection Reagent (Promega # 2691) overnight. The Opti-MEM media of 293T cell culture was changed to HBMEC complete media, and cells were incubated overnight. HBMEC were infected by lentivirus in the 0.22 μm filtered 293T cell culture supernatant for 48–72 h and the shRNA knockdown HBMECs were selected with HBMEC complete media supplemented with 1 μg/ml puromycin. The expression of the CysLT1 and ezrin was verified by Western blotting.

## Mouse model of experimental hematogenous meningitis

Each mouse received $1 \times 10^7$ CFUs of GBS strain K79 in 100 μl PBS via the tail vein. One hour later, mouse chest was cut open, and blood from right ventricle was collected and plated for bacterial counts, which were expressed as CFUs/ml of blood. The mouse was then perfused with a mammalian Ringer's solution by transcardiac perfusion through a 23-gauge needle inserted into the left ventricle of the heart under the perfusion pressure of about 100 mmHg as previously described (Zhu et al, 2010; Maruvada et al, 2011). The brains were removed, weighed, and homogenized in 2 ml RPMI followed by plating for bacterial counts, which were expressed as CFUs/gm. In some experiments, montelukast and gefitinib, applied at therapeutic doses (5 mg/kg for montelukast, and 10 mg/kg for gefitinib) (Wang et al, 2009; Wang et al, 2016; Zhu et al, 2017), were intraperitoneally administrated 1 h before bacterial challenge. For therapy of established GBS meningitis, animals received GBS via the tail vein, and 24 h later, received ceftriaxone (100 mg/kg), montelukast (10 mg/kg), or combination daily for 7 days.

## Examination of entry sites for circulating GBS penetration into the brain and assessment of the blood–brain barrier permeability

Each mouse received $1 \times 10^7$ CFUs of GFP-K79 in 100 μl PBS via the tail vein as described above. At 1 and 12 h later, animals were perfused with PBS followed by 2% paraformaldehyde (PFA). The brains and skullcap were removed, fixed overnight with 2% PFA at 4°C, and then re-hydrated in 1× PBS at 4°C for 3 h. The brains were embedded in 3% agarose and cut into sections of 150 μm thickness using a vibratome (Leica). The meninges were carefully detached from the skullcap. To assess the blood–brain barrier permeability with GBS penetration into the brain, 200 μl solution (20 mg/ml) of Sulfo-NHS-biotin (a low molecular weight tracer with m.w. of 443 Da) was intraperitoneally injected 10 min before intracardiac perfusion. Covalently, immobilized Sulfo-NHS-biotin was visualized with fluorescent streptavidin, and extravasation is indicative of leakage from the brain microvessels. The brain sections and meninges were incubated with CD31 antibody (a marker for capillaries) diluted in 1 × PBSTC (1 × PBS + 1% Triton X-100 + 0.1 mM CaCl$_2$) + 5% BSA overnight at 4°C. Then, they were washed 3 times with 1 × PBSTC and subsequently incubated overnight with different secondary antibodies or Texas Red streptavidin diluted in

1 × PBSTC + 5% BSA. The next day, the brain sections and meninges were washed 3 times with 1 × PBSTC, and flat-mounted using ProLong Gold antifade reagent (Invitrogen). The tissues were imaged using a Zeiss LSM700 confocal microscope and processed with ImageJ (Wang *et al*, 2019).

## Examination of structural integrity of the blood–brain barrier during GBS penetration

The mice received $1 \times 10^7$ CFUs of GFP-K79 in 100 μl PBS via the tail vein as described above. One hour later, the mice were perfused with PBS and 2% PFA. The brain sections were stained with claudin-5 antibody (a marker for tight junction proteins) diluted in 1 × PBSTC + 5% BSA overnight. After washing three times, the sections were incubated with 568-conjugated secondary antibody and assessed by the Zeiss LSM700 confocal microscope. The 2D figures and 3D figures were generated by Zen software and analyzed by ImageJ.

## Immunofluorescence in HBMEC

Control and S1P2 knockout HBMEC were grown on collagen-coated glass slide to confluence and incubated with GFP-K79 at MOI of 1:100 at 37°C for 90 min. Cells were washed with PBS to remove the free, unbound bacteria, and then fixed with 4% paraformaldehyde. The cells were incubated with rabbit anti-GBS antiserum (1:500) diluted in PBS for 30 min at room temperature to distinguish intracellular and extracellular GBS. Then, the cells were washed by PBS and permeabilized with 0.25% Triton X-100 solution for 10 min, and blocked with 5% BSA for 1 h. The cells were then incubated with EGFR or Ezrin monoclonal antibody and subsequently incubated with anti-mouse Alexa Fluor-488 and anti-rabbit Fluor-568 labeled secondary antibody. The glass slide was mounted and visualized using fluorescence microscopy as previously described (Wang *et al*, 2016). Percentage of co-localization was calculated by counting numbers of all GBS and co-localized intracellular GBS from at least three representative fields, and expressing as numbers of co-localized GBS/all GBS × 100. All data correspond to the mean ± SEM of at least 60 GBS from three fields.

## Histopathology assessment of neuronal injury

Animals were anesthetized with an intraperitoneal administration of 100 μl dose per mouse of cocktail of ketamine (87.5 mg/kg) and xylazine (12.5 mg/kg) and perfused transcardially with a phosphate-buffered saline, and the brains were fixed with fixing solution (4% PFA, in 0.1 M PBS, pH 7.2) through infusion pump. Brains were post-fixed in 4% PFA at 4°C for 24 h and cryoprotected in 35% sucrose (w/v) in PBS for at least 2 days at 4°C. Fixed brains were cut into 20 micrometer thick coronal slices on gelatin-coated slides by cryostat (Leica).

## Nissl staining

Sections were then defatted for 10 min in a mixture of methanol/acetone (1:1, vol). Slices were dehydrated in a series of ethanol washes (70–100%). Sections were immersed in 1% nissl stain dye

for 5 s then rinsed in dH2O, followed by a series of graded alcohols, cleared in xylene, and coverslipped with a mounting medium. Images were acquired on a Zeiss LSM microscope. Neurons with only healthy morphological characteristics of nucleus were counted in the cortex region of the brain section. Data were obtained from six fields per slide, and the average numbers of cells in each hemisphere, in five sections from each animal was analyzed for quantification, and the mean was used for the statistical analysis. Graph results are expressed as the number of Nissl bodies (Del Toro *et al*, 2017).

## Apoptosis assay

The extent of apoptosis TUNEL assay (TdT-mediated dUTP-X nick end labeling) from all the groups was analyzed using an in situ cell death detection kit (Roche), according to the manufacturer's instructions and counterstained with DAPI for nuclei in the cryosection of brain slide. The images were taken using a fluorescence microscope (Zeiss LSM microscope). The mean TUNEL-positive nuclei will be counted from three different sections per animal for quantification.

## Immunohistochemistry of microglia and astrocytes

Brain sections were immunostained by first blocking non-specific binding sites with 2% IgG-free bovine serum albumin (BSA, Sigma) and then incubating the tissue with 0.2% Triton X-100. The sections were then incubated overnight at 4°C with a primary antibody against microglia marker (Iba-1, 1:2,000, Abcam) and glial fibrillary acidic protein (GFAP 1:500, Dako, Denmark) followed by fluorescence labeled secondary antibody Alexa Fluor 647 and 488 (Invitrogen). The nuclei were counterstained with VectaShield DAPI (Vector Labs), and photomicrographs of cortical and hippocampal subfields were obtained on a fluorescence microscope (Zeiss). In the microphotomicrographs, the number of cells immunoreactive to Iba-1 and GFAP were quantified in the different region of cortex and hippocampus (Chen *et al*, 2015).

## Assessment of memory function following GBS meningitis using Y-maze

A Y-shaped white-painted timber with arms 29.5 cm long × 7.5 cm wide × 15.5 cm high was used. All mice were subjected to a 2-trial Y-maze test separated by a 24 h intertrial interval to assess spatial recognition memory, with all testing performed during the light phase of the circadian cycle. The 3 identical arms were randomly designated start arm, novel arm, and other arm. Visual cues were placed on the walls of the maze. The first trial (training) was for 10 min, and the mice were allowed to explore only 2 arms (starting arm and other arm). For the second trial (retention), mice were placed back in the maze in the same starting arm and allowed to explore for 5 min with free access to all 3 arms. Behaviors were recorded on video during a 5-min trial, and the Ethovision video-tracking system was used for analysis. Data are expressed as the total duration of novel arm and percentage of entries in novel arm made during the 5-min second trial. Calculation of percentage of entries in novel arm is as follows: entry of novel arm/total entry (novel + start + other) × 100 (Zhao *et al*, 2018).

**The paper explained**

**Problem**

GBS meningitis continues to be an important cause of mortality and morbidity. GBS strains are uniformly susceptible to penicillin and poor outcome of GBS meningitis is not due to emergence of non-susceptible GBS strains. At present, no strategy exists for development of improved therapy.

**Results**

Our findings reveal that development of an adjunct therapy for improving the outcome of GBS meningitis is feasible by targeting host factors involved in GBS invasion of the blood–brain barrier.

**Impact**

Determination of targets as an adjunct therapy of GBS meningitis is a breakthrough concept for GBS meningitis, encouraging continued investigation of host factors involved in GBS invasion of the blood–brain barrier.

## Ethics statement

Our animal studies were carried out in strict accordance with the current recommendations in the Guide for the Care and Use of Handling Animals, NIH publication DHHS/USPHS. Numbers of animals for experimental groups were determined by power analyses based on our preliminary data. The animal protocol was approved by The Johns Hopkins Animal Care and Use Committee (Animal Welfare Assurance Number: A3272-01). All efforts were made to provide the ethical treatment and minimize suffering of animals employed in this study.

## Statistical analysis

Data are expressed as mean $\pm$ SEM. Differences of bacterial counts in the brain (CFUs/gm) between different groups of mice were determined by Wilcoxon rank sum test or Student's $t$-test. Differences of bacterial invasion in HBMEC monolayer were determined by Student's $t$-test. Survival curves were generated using the Kaplan–Meier method, and differences were assessed by a two-sided log-rank (Mantel-Cox) test (GraphPad software, version 6.0). One-way ANOVA followed by multiple comparison of Bonferroni's post hoc test was used to determine differences of quantification of image analysis, morbidity, and motor activity scores as well as Y-maze data. $P < 0.05$ was considered significant. N-values for experimental groups were determined by power analyses based on our preliminary data. All *in vivo* analyses were performed in a double-blind manner.

## Data availability

- The datasets produced in this study are available in the following databases: RNA sequencing: NCBI sequence read archive (SRA), BioProject accession no. PRJNA632824 (https://www.ncbi.nlm.nih.gov/bioproject/PRJNA632824/).
- All data necessary to understand and evaluate the conclusion of this report are available in the paper and/or supplementary materials.

Expanded View for this article is available online.

## Acknowledgements

We thank D. Pearce for technical assistance in experiments with HBMECs, Vincent Bruno for RNA-seq experiments, and K. Frank Austen and Yoshihide Kanaoka for providing CysLT1$^{-/-}$ and CysLT2$^{-/-}$ mice. This work was supported by the US National Institutes of Health (NIH) grants, NS091102, AI84984, AI113273, and AI126176 to KSK.

## Author contributions

CZ, ZH, NZ, WL, AP, and WS performed the experiments. AM performed S1P measurements. KSK conceived the project and wrote the paper, and all authors edited the manuscript.

## Conflict of interest

The authors declare that they have no conflict of interest.

## For more information

- https://www.hopkinsmedicine.org/profiles/results/directory/profile/0015326/kwang-kim
- https://www.jhsph.edu/faculty/directory/profile/963/kwang-sik-kim
  The authors' websites.

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
