## [Review Process File · EMBO Molecular Medicine]

Therapeutic development of group B *Streptococcus* meningitis by targeting a host cell signaling network involving EGFR

Ningyu Zhu, Chengxian Zhang, Zheng Hou, Atish Prakash, Wei Liu, Weifeng She, Andrew Morris, and Kwang Kim

DOI: [10.15252/emmm.202012651](https://doi.org/10.15252/emmm.202012651)

Corresponding authors: Kwang Kim (kwangkim@jhmi.edu)

Review Timeline:

Submission Date:	1st May 20
Editorial Decision:	3rd Jun 20
Revision Received:	8th Oct 20
Editorial Decision:	11th Nov 20
Revision Received:	10th Dec 20
Accepted:	11th Dec 20

Editor: Zeljko Durdevic

Transaction Report:

3rd Jun 2020

Dear Prof. Kim,

Thank you for the submission of your manuscript to EMBO Molecular Medicine. We have now heard back from the three referees who agreed to evaluate your manuscript. As you will see from the reports below, the referees acknowledge the interest of the study. However, they raise serious concerns that should be addressed in a major revision of the present manuscript. For further consideration of the manuscript it will be essential to better define the molecular mechanism for the GBS interaction with the HBMECs, to demonstrate transcellular passage of the GBS using a transwell assay, and to show that the combination of inhibitors in addition to the antibiotics further increases survival of infected mice. Addressing the reviewers' concerns in full will be necessary for further considering the manuscript in our journal.

Acceptance of the manuscript will entail a second round of review. Please note that EMBO Molecular Medicine encourages a single round of revision only and therefore, acceptance or rejection of the manuscript will depend on the completeness of your responses included in the next, final version of the manuscript. For this reason, and to save you from any frustrations in the end, I would strongly advise against returning an incomplete revision.

We realize that the current situation is exceptional on the account of the COVID-19/SARS-CoV-2 pandemic. Therefore, please let us know if you need more than three months to revise the manuscript.

I look forward to receiving your revised manuscript.

***** Reviewer's comments *****

Referee #1 (Remarks for Author):

General comments:

The authors performed a very comprehensive investigation into the GBS invasion mechanism during bacterial meningitis. They were able to identify EGFR as a central player in GBS invasion and could show that GBS uses a transcellular route. Through specific inhibition of EGFR as well as upstream and downstream signaling molecules the authors could show the involvement of the host factors EGFR, SIP and CysLT1 in cell culture as well as in animal experiments. Furthermore they

were able to show the potential of the CysLT1 inhibitor Montelukast and the EGFR inhibitor Gefitinib for the development of innovative therapeutic strategies for treating GBS meningitis. To demonstrate the specificity of the inhibitors their effect on GBS invasion should be studied in CysLT1 or EGFR knock out cells.

Specific comments:

1 While I am not a native speaker to me the title sounds really confusing. I had to read the abstract, to understand it. The authors should consider to modify it, so that it becomes clear that interference with EGFR signaling pathways may have a therapeutic potential for GBS meningitis.

2 Line 118 the findings of this manuscript point to meningeal and cortex capillaries as relevant for GBS invasion of the CNS. However in typical cases GBS causes meningitis and not multiple brains abscesses in the cortex, which would be expected if cortex capillaries are the relevant cells for GBS invasion. It is a point that should be discussed in more detail.

3 line 145-147

To demonstrate the specificity of the EGFR inhibitor Gefitinib, the authors should investigate the effect of Gefitinib in EGFR knock out cells. If it only affects GBS invasion through inhibition of EGFR it should have no effect on GBS invasion in EGFR knock out cells.

4 line 218-220 see comment above

To demonstrate the specificity of the CysLT1 inhibitor Montelukast, the authors should investigate the effect of Montelukast in CysLT1 knock out cells. If it only affects GBS invasion through inhibition of CysLT1 it should have no effect on GBS invasion in CysLT1 knock out cells.

5 line 341-343 Discussion

The authors should discuss the role of EGFR in E. coli meningitis and the differences and similarities between the involvement of EGFR in E. coli and GBS meningitis in more detail.

6 The figure legend of Fig. 3 states: (C, E, F) Bacterial counts recovered from the blood and brain in wild type mice, CysLT1 $-/-$ mice, CysLT2 $-/-$ mice, wild type mice receiving vehicle control or montelukast (5 mg/kg), infected with strain K79 for 1h (mean {plus minus} SEM).

While the wildtype strains cause bacteremia at a level of 4 cfu/ml of blood in all three experiments cfu in brain tissue differ a lot. The wt control mice in the Montelukast experiment have a level of 2500 cfu/g of brain tissue while in in the other two experiments the wt control mice have a level of about 600-700 cfu/g of brain tissue. The difference in the control animals is strange, please explain, why in the controls of the Montelukast experiment the cfus in brain tissue are so much higher. According to Materials and Methods all of the control animals were infected with the same amount of GBS.

Referee #2 (Remarks for Author):

In their study, "Therapeutic development of group B Streptococcus meningitis by exploring a host cell network" (EMM-2020-12651), Zhu et al study the penetration of group B Streptococcus (GBS) into the brain by way of a network that is comprised of S1P2, EGFR and CysLT1. Using pharmacologic inhibition, gene knockout and knockdown cells as well as gene knockout animals, they conclude that targeting this network therapeutically improves the outcome of animals with GBS meningitis.

This study addresses how the GBS penetrates the blood-brain barrier. Citing several studies suggesting that this occurs in cerebral microvessels, they note this may occur via transcellular, paracellular or via "hijacking" mechanisms. The investigators use human brain microvascular endothelial cells (HBMECs) to show that GBS invades these cells without accompanying immune cells, change in blood brain barrier (BBB) permeability as evaluated by no change in the electric resistance (TEER), cell viability or allowing penetration of a small molecule, biotin, into the animal brain and therefore conclude the most likely mode of penetration of the BBB is it most likely by transcellular penetration. They further define the mechanism using transcriptome analysis of HBMEC (RNA-seq) to define the involvement of EGFR, sphingosine 1-phosphate (S1P), cytosolic phospholipase 2a (cPLA2a), cysteinyl leukotrienes (CysLT) and ezrin-radix-moesin (ERM). They show that loss of function of these specific host factors by either inhibitors or gene deletion, both in vitro and in vivo improves outcome of GBS meningitis.

Major Comments

This study uses multiple state-of-the-art techniques, both in vivo and in vitro, and analytic methods to support their findings that each of the components of this network are critical to GBS meningitis, with EGFR playing a leading role. While they show in brain sections that some of the intravenously injected GBS can be found outside the meningeal and cortex capillaries, indicative of successful penetration of the BBB, it is not clear why they did not show penetration of GBS across the HBMEC in a Transwell assay rather than HBMEC invasion only. Thus the only support for their inference that GBS penetrates the BBB by translocation is the finding of a "few bacteria" outside these capillaries, the absence of a small molecule tracer into the brain and intact TEER and cell viability. They show that GBS can enter the HBMEC but do not demonstrate in this in vitro assay if they get out. Are we to assume that the same mechanisms that permit GBS invasion of HBMECs also are operative in their release from these cells into the brain? Are there any EMs showing intact GBS leaving the basal side of the cells? They do show that inhibition of components of this pathway does lead to marked decreases in the GBS in the brain while not affecting the GBS levels in the blood, but this finding does not unequivocally imply that they arrived in the brain by translocation. They show that inhibitors of each of the components of this network block HBMEC invasion both in vivo and in vitro and by these methods conclude that each of these components is involved in BBB penetration. They go on to establish the principle that a combination of antibiotics and immunologic modification can improve outcome of GBS meningitis in mice.

They find that the contributions of S1P2, EGFR and cPLA2a-CysLT1-ezrin to GBS penetration of the BBB were inter-related and propose a model (Fig. 3J). I am unaware of any bacterial pathogen that utilizes such a complicated mechanism for cell invasion. Are there other precedents? For one, how does GBS initiate this cascade? Are the bacteria endocytosed, and if so, does inhibition of endocytosis block the cascade? Do the GBS activate the endothelial cell? Do GBS bind to any of these components or any other cell surface receptor? How does the EGFR become activated-it must dimerize to acquire function. They do suggest there is "inside-out" activation of EGFR. How often does that occur?

Specific Comments

Figure 2B. If the SphK1/2 inhibitors decrease invasion, is involvement of EGFR necessary?

Fig3C. Are the differences significant?

Fig 4A. This is the only survival curve shown for any network component. Does inhibition of each component lead to similar survival curves? Will addition of two inhibitors to the antibiotic improve survival even further? i.e. will blockade of two components in the putative sequential network provide even greater survival?

Their model postulates at least "inside-out" activation of two cell surface receptors (S1P2 and CysLT1) in series. Again, is this unusual?

Might ERM that is involved in host cell actin cytoskeleton rearrangements play a role in the initial endocytosis of GBS by the HBMEC?

L292 What type of clinical trial has the CysLT1 been involved?

L. 345. Might the authors speculate why a high degree of bacteremia is required for BBB invasion. Is there a dose dependency to the invasion of GBS in their in vitro models and is this related to the number of colonies required to activate the network?

L366 The authors use the argument that knockout of S1P2 led to the inhibition of EGFR and as support for the inter-relatedness of this network. But in Fig 2D they show a profound decrease in GBS CFU presumably without involvement of EGFR, and of further downstream host factors. In many of the figures the concentration of the inhibitors or antibiotic are not clearly indicated.

Referee #3 (Remarks for Author):

In this manuscript, Zhu et al. report that GBS, the first etiologic agent for neonatal meningitis, exploits a signalling platform involving SphK1/2, S1P, S1P2 receptor, EGFR, PLA2 α , CysLT1 for penetration of the blood brain barrier by a transcellular mechanism. Blocking signalling molecules from this platform decreased GBS crossing into the brain in an in vivo model of meningitis and improved the outcome of the animals.

Major comments:

Molecular mechanisms for GBS penetration of the BBB are lacking. However, although results from this paper contribute to a better understanding of mechanisms used by GBS to cross the brain barrier, the novelty of the overall finding of this study is not so attractive for several important reasons:

1- SphK1/2, S1P, S1P2 receptor, EGFR, signalling pathway has already been described by the same group to be involved in BBB crossing by *Escherichia coli* K1, the other pathogen responsible for meningitis in neonates (Wang et al. PLoS pathogen 2016). More recently, the authors completed this signalling platform by the identification of PLA2 α and CysLTs downstream EGFR (Zhu et al. Cell Microbiol. 2020). Although it is very interesting that these 2 unrelated pathogens (Gram negative bacilli versus Gram positive coccus) use the same signalling pathway, identifying such pathway for another pathogen than *E. coli* K1 has a limited impact. The Wang et al. paper is referenced in the manuscript, but exclusively as a technical reference. Zhu et al. cell microbiol. 2020 paper is not referenced, the current paper seeming consequently more original than it is. The involvement of this signalling pathway that seems to be common for these 2 pathogens is not discussed in the manuscript. This point is of particular importance as it raises the question of GBS virulence factors involved in this platform activation. Indeed, in *E. coli* K1, OmpA, FimH and Nlp1 are the virulence factors responsible for S1P, EGFR, PLA2 α , CysLT1 activation (Wang et al. PLoS pathogen 2016) while those virulence factors are not expressed by GBS. Therefore, what are GBS virulence factors involved in the activation of this signalling pathway?

Also, for *E. coli* K1, c-Src was identified as the signalling molecule between S1P2 and EGFR? Is it

also the case for GBS?

2- In addition to SphK1/2, S1P, S1P2 receptor, EGFR, signalling pathway, the authors further decipher the signalling cascade that also involves PLA2 α and CysLT1. As mentioned by the authors, the involvement of PLA2 α , CysLT1 in BBB crossing by GBS has been published previously by this group (Maruvada et al. *IAI* 2011). The involvement of CysLT1 in GBS invasion was demonstrated using Montelukast. It was also known that Montelukast inhibits GBS penetration of the BBB in vivo (Syu et al. *Nature com.* 2019). Because Fig. 3 B, C, D, E, F only confirmed previously published results, it should not appear in the results but as data not shown or supplementary information.

3- The very interesting point in the characterization of this signalling platform is that SphK1/2, S1P, S1P2 receptor, EGFR, signalling is connected to PLA2 α , CysLT1 signaling. However, data of this interconnection are not that convincing. The difference in phosphorylation status of PLA2 α in control versus EGFR KO (Fig. 3A right panel) or S1P2 KO (Fig. 3J) are very weak and the uninfected condition is not shown. The reviewer is not convinced by images of EGFR and ezrin recruitment around GBS associated cells (FIG 3L).

4- Fig. 4 shows that Montelukast that acts as an antagonist of the CystLT1 receptor (at the end of the signalling pathway) in adjunct treatment with ceftriaxone strongly improves the outcome of infected mice. This result is particularly important and convincing. This part, that is central to the manuscript, should be further studied. Does gefitinib that act upstream also improves mice outcome? Are there any additional effects on survival if gefitinib and Montelukast are given together with ceftriaxone?

5- Authors claim (lanes 122-124) that GBS penetration into the brain occurs in the meningeal and cortex capillaries by a transcellular mechanism. However, the data presented do not support this statement. No GFP bacteria are visible in fig 1B and Fig. S2 A, B and C (contrast problem or image format?). Hence, the reviewer cannot say that bacteria are visible outside of capillaries at early time points in cortex and meningeal capillaries while they are absent from choroid plexuses. In addition, the transcellular mechanism is not demonstrated. Indeed, authors claim it is a transcellular passage as bacteria can be found outside the capillaries while there was no extravasation of intravascular tracer (sulfo-NHS-Biotin). However, several groups fail to observe the diffusion of tracers during diapedesis of immune cells by paracellular mechanism at the BBB demonstrating that barrier function can be maintained during paracellular diapedesis of immune cells at the BBB (Winger et al. *J Immunol* 2014; Engelhardt et al. *European Journal of immunology* 2004...). The lack of tracer diffusion is therefore not sufficient to assert that GBS does not cross BBB by a paracellular mechanism.

The authors could assay GBS transcellular passage by performing transcellular migration assay on an appropriate cell line, such as HBMEC, in the presence of gefitinib or in EGFR KO cells.

Minor comments:

Fig. S2A time point 12hrs, capillary and GFP bacteria pictures have been inverted

Fig. 1A: some letters are truncated on the right of the figure.

Lane 189: should it not be Fig. S3 instead of Fig. S2A?

Some experimental information is missing (origin of GFP-GBS strain? is GFP expression constitutive or inducible, chromosomal or plasmid encoded?); which choroid plexus have been used in figure 1B and Fig. S2 C (lateral, 3rd, or 4th ventricle)?

Two different statistical tests described in material and methods (Wilcoxon and student) have been used for in vivo experiments. The reader should be informed of which test has been used for each

figure. Also, for invasion experiments displaying more than 2 sets of data, the ANOVA test to correct for multiple comparisons would be more appropriate.

Experiments have been performed in triplicate but authors should indicate the number of independent experiments that have been realized.

Referee #1 (Remarks for Author):

General comments:

The authors performed a very comprehensive investigation into the GBS invasion mechanism during bacterial meningitis. They were able to identify EGFR as a central player in GBS invasion and could show that GBS uses a transcellular route. Through specific inhibition of EGFR as well as upstream and downstream signaling molecules the authors could show the involvement of the host factors EGFR, SIP and CysLT1 in cell culture as well as in animal experiments. Furthermore they were able to show the potential of the CysLT1 inhibitor Montelukast and the EGFR inhibitor Gefitinib for the development of innovative therapeutic strategies for treating GBS meningitis. To demonstrate the specificity of the inhibitors their effect on GBS invasion should be studied in CysLT1 or EGFR knock out cells.

Specific comments:

1 While I am not a native speaker to me the title sounds really confusing. I had to read the abstract, to understand it. The authors should consider to modify it, so that it becomes clear that interference with EGFR signaling pathways may have a therapeutic potential for GBS meningitis. – **the title was modified as follows. Therapeutic development of group B *Streptococcus meningitis* by targeting a host cell signaling network involving EGFR**

2 Line 118 the findings of this manuscript point to meningeal and cortex capillaries as relevant for GBS invasion of the CNS. However in typical cases GBS causes meningitis and not multiple brains abscesses in the cortex, which would be expected if cortex capillaries are the relevant cells for GBS invasion. It is a point that should be discussed in more detail. – **line 120 This point was discussed in more detail, demonstrating that GBS invasion of the CNS is a continuous process involving initially meningeal and subsequently cortex capillaries, and clinical cases of GBS meningitis presenting with cerebritis illustrate this point (Kim KS, et al. Cerebritis due to group B streptococcus. Scan J Infect Dis 14:305-308, 1982). In addition, our experimental GBS meningitis model has been used by other investigators for investigating the pathogenesis of GBS meningitis (Doran et al. JCI 2005 10.1172/JCI23829; Banerjee et al. Nature Commun. 2015 doi: 10.1038/ncomms1474; Deng et al. Plos Pathogen 2019 <https://doi.org/10.1371/journal.ppat.1007848>).**

3 line 145-147 To demonstrate the specificity of the EGFR inhibitor Gefitinib, the authors should investigate the effect of Gefitinib in EGFR knock out cells. If it only affects GBS invasion through inhibition of EGFR it should have no effect on GBS invasion in EGFR knock out cells. – **line 158 The specificity of Gefitinib for the effect on GBS invasion in EGFR knockout cells is provided, demonstrating no effect of gefitinib on GBS invasion in EGFR knockout cells (Fig. S4A)**

4 line 218-220 see comment above. To demonstrate the specificity of the CysLT1 inhibitor Montelukast, the authors should investigate the effect of Montelukast in CysLT1 knock out cells. If it only affects GBS invasion through inhibition of CysLT1 it should have no effect on GBS invasion in CysLT1 knock out cells. –line 234 The specificity of Montelukast for the effect on GBS invasion in CysLT1 knockdown cells is provided, demonstrating no effect of montelukast on GBS invasion in CysLT1 knockdown cells (Fig. S4A)

5 line 341-343 Discussion

The authors should discuss the role of EGFR in *E. coli* meningitis and the differences and similarities between the involvement of EGFR in *E. coli* and GBS meningitis in more detail. – differences and similarities between *E. coli* and GBS are provided on line 417 – The

contribution of EGFR-CysLT1 network to GBS invasion differs from that of *E. coli* invasion. For example, host cell signaling pathway exploited by *E. coli* invasion involves both the EGFR-CysLT1-dependent and independent pathways, as shown by the demonstration that decreased *E. coli* invasion in EGFR and CysLT1 knockout/knockdown HBMEC can be further reduced by montelukast and gefitinib, respectively. Strain RS218 is a meningitis isolate of *E. coli*.

In contrast, GBS invasion was entirely dependent upon the EGFR-CysLT1 pathway, as shown by no additional inhibition of GBS invasion by montelukast and gefitinib in EGFR and CysLT1 knockout/knockdown HBMEC, respectively. This issue was discussed on line 417 (Fig. S4A and 4C).

6 The figure legend of Fig. 3 states: (C, E, F) Bacterial counts recovered from the blood and

brain in wild type mice, CysLT1 ^{-/-} mice, CysLT2^{-/-} mice, wild type mice receiving vehicle control or montelukast (5 mg/kg), infected with strain K79 for 1h (mean {plus minus} SEM).

While the wildtype strains cause bacteremia at a level of 4 cfu/ml of blood in all three experiments cfu in brain tissue differ a lot. The wt control mice in the Montelukast experiment have a level of 2500 cfu/g of brain tissue while in in the other two experiments the wt control mice have a level of about 600-700 cfu/g of brain tissue. The difference in the control animals is strange, please explain, why in the controls of the Montelukast experiment the cfus in brain tissue are so much higher. According to Materials and Methods all of the control animals were infected with the same amount of GBS.- **As this reviewer is aware of, there are inter-litter and inter-animal variations for any animal experiments and it is, therefore, important to include control groups in every animals experiments.**

Referee #2 (Remarks for Author):

In their study, "Therapeutic development of group B Streptococcus meningitis by exploring a host cell network" (EMM-2020-12651), Zhu et al study the penetration of group B Streptococcus (GBS) into the brain by way of a network that is comprised of S1P2, EGFR and CysLT1. Using pharmacologic inhibition, gene knockout and knockdown cells as well as gene knockout animals, they conclude that targeting this network therapeutically improves the outcome of animals with GBS meningitis.

This study addresses how the GBS penetrates the blood-brain barrier. Citing several studies suggesting that this occurs in cerebral microvessels, they note this may occur via transcellular, paracellular or via "hijacking" mechanisms. The investigators use human brain microvascular endothelial cells (HBMECs) to show that GBS invades these cells without accompanying immune cells, change in blood brain barrier (BBB) permeability as evaluated by no change in the electric resistance (TEER), cell viability or allowing penetration of a small molecule, biotin, into the animal brain and therefore conclude the most likely mode of penetration of the BBB is it most likely by transcellular penetration. They further define the mechanism using transcriptome analysis of HBMEC (RNA-seq) to define the involvement of EGFR, sphingosine 1-phosphate (S1P), cytosolic phospholipase 2a (cPLA2a), cysteinyl leukotrienes (CysLT) and ezrin-radix-moesin (ERM). They show that loss of function of these specific host factors by either inhibitors or gene deletion, both in vitro and in vivo improves outcome of GBS meningitis.

Major Comments

This study uses multiple state-of-the art techniques, both in vivo and in vitro, and analytic methods to support their findings that each of the components of this network are critical to GBS meningitis, with EGFR playing a leading role. While they show in brain sections that some of the intravenously injected GBS can be found outside the meningeal and cortex capillaries, indicative of successful penetration of the BBB, it is not clear why they did not show penetration of GBS across the HBMEC in a Transwell assay rather than HBMEC invasion only. Thus the only support for their inference that GBS penetrates the BBB by translocation is the finding of a "few bacteria" outside these capillaries, the absence of a small molecule tracer into the brain and intact TEER and cell viability. They show that GBS can enter the HBMEC but do not demonstrate in

this in vitro assay if they get out.- line 158, 205 and 250 data on GBS penetration across HBMEC monolayer on Transwell assays are provided (Fig. 1H, 2D and 3E), demonstrating that GBS penetration across the HBMEC monolayer is decreased in EGFR and S1P2 knockout cells and also in CysLT1 knockdown cells compared to their respective control HBMEC .

p<0.01, *p<0.001

Are we to assume that the same mechanisms that permit GBS invasion of HBMECs also are operative in their release from these cells into the brain? Are there any EMs showing intact GBS leaving the basal side of the cells? – Transmission EM showing a transcellular GBS penetration of HBMEC was published (Nizet et al 1997). It is, however, unknown how GBS is released from the HBMEC basal side. Based on information available with other CNS-infecting pathogens (e.g., *E. coli*) it is likely that the mechanism involved in GBS exocytosis differs from GBS invasion of HBMEC and this will be the topic of future investigation.

They do show that inhibition of components of this pathway does lead to marked decreases in the GBS in the brain while not affecting the GBS levels in the blood, but this finding does not unequivocally imply that they arrived in the brain by translocation. They show that inhibitors of each of the components of this network block HBMEC invasion both in vivo and in vitro and by these methods conclude that each of these components is involved in BBB penetration. They go on to establish the principle that a combination of antibiotics and immunologic modification can improve outcome of GBS meningitis in mice. They find that the contributions of S1P2, EGFR and cPLA2a-CysLT1-ezrin to GBS penetration of the BBB were inter-related and propose a model (Fig. 3J). I am unaware of any bacterial pathogen that utilizes such a complicated mechanism for cell invasion. Are there other precedents? – We believe that this complicated mechanism is a unique trait for CNS-infecting pathogens to invade the BBB

For one, how does GBS initiate this cascade? Are the bacteria endocytosed, and if so, does inhibition of endocytosis block the cascade? Do the GBS activate the endothelial cell? Do GBS bind to any of these components or any other cell surface receptor? How does the EGFR become activated-it must dimerize to acquire function. They do suggest there is "inside-out" activation of EGFR. How often does that occur?

We showed that GBS invasion of HBMEC involves clathrin-mediated endocytosis and inhibition of endocytosis was effective in inhibition of GBS invasion. This concept is shown with dynasore (a cell-permeable inhibitor of clathrin-mediated endocytosis, Veiga et al 2007), and dynasore inhibited GBS invasion of HBMEC. These findings support the role of endocytosis in GBS invasion of HBMEC. This concept is also supported by the demonstration that EGFR activation

in response to GBS was inhibited by dynasore, supporting that inhibition of endocytosis results in inhibition of host cell signaling cascade involved in GBS invasion of HBMEC (Fig. S4D).

In addition, our findings showed EGFR transactivation in response to GBS depends on S1P-S1P2, as shown by the demonstration that inhibition and deletion of S1P2 inhibited EGFR activation in response to GBS (Fig. 2A). We also showed that cPLA2 α activation in response to GBS depends on S1P2-EGFR-CysLT1, as shown by the demonstration that gene knockout of S1P2 and EGFR and knockdown of CysLT1 inhibited cPLA2 α activation (Fig. 3B). In addition, we showed that co-localization of intracellular GBS with EGFR and ezrin is decreased in S1P2 knockout HBMEC.

Specific Comments

Figure 2B. If the SphK1/2 inhibitors decrease invasion, is involvement of EGFR necessary? – based on our data demonstrating no additional inhibition by combining SIP2 and EGFR, it is not likely to be necessary and EGFR functions as downstream of SIP2 in GBS invasion of the blood-brain barrier.

Fig3C. Are the differences significant? – the differences in bacterial counts recovered from the brains of WT and CysLT2^{-/-} mice were not significant.

Fig 4A. This is the only survival curve shown for any network component. Does inhibition of each component lead to similar survival curves? Will addition of two inhibitors to the antibiotic improve survival even further? i.e. will blockade of two components in the putative sequential network provide even greater survival? – We have shown that GBS invasion of the blood-brain barrier is dependent on the EGFR-CysLT1 pathway and there was no additive effect on GBS invasion with inhibition of both EGFR and CysLT1, as shown by the demonstration that no additional inhibition of GBS invasion with gefitinib in CysLT1 KD cells and no additional inhibition with montelukast in EGFR KO cells (Fig. S4A).

We also examined the efficacy of ceftriaxone plus gefitinib as well as ceftriaxone plus gefitinib and montelukast. As shown in (Fig. S4B), survival was greater with ceftriaxone plus gefitinib or montelukast compared to ceftriaxone alone. However, administration of two inhibitors together did not provide additional improved survival compared to individual inhibitor (line 443). These findings are consistent with our in vitro finding that inhibition of both EGFR and CysLT1 does not exhibit any additive inhibition compared to individual inhibitors.

* $p < 0.05$ between GBS+ceftriaxone vs GBS+ceftriaxone+montelukast, GBS+ceftriaxone+gefitinib and GBS+ceftriaxone+montelukast+gefitinib by a log-rank test.

Their model postulates at least "inside-out" activation of two cell surface receptors (S1P2 and CysLT1) in series. Again, is this unusual? - **This is a very unique scenario for GBS exploitation of S1P2-EGFR-CysLT1 for penetration of the blood-brain barrier.**

Might ERM that is involved in host cell actin cytoskeleton rearrangements play a role in the initial endocytosis of GBS by the HBMEC? - **Based on our data, ERM involvement is likely to occur after CysLT1, as shown by the demonstration that ezrin activation in response to GBS was inhibited in CysLT1^{-/-} mice (Fig. 3I).**

L.292 What type of clinical trial has the CysLT1 been involved? - **CysLT1 antagonists have been used in clinical studies of respiratory disorders such as asthma (Kemp JP 2005).**

L. 345. Might the authors speculate why a high degree of bacteremia is required for BBB invasion. Is there a dose dependency to the invasion of GBS in their in vitro models and is this related to the number of colonies required to activate the network? -**line 378 the mechanisms involved with a high-degree of bacteremia in GBS invasion of the BBB remain incompletely understood. Our data showed that EGFR activation in response to GBS was inoculum-dependent (shown here, EGFR activation with a multiplicity of infection of 100, not with 1), suggesting that one mechanism requiring a high-degree of bacteremia for BBB invasion is related to dose-dependent activation of host cell signaling molecules involved in GBS invasion of the blood-brain barrier (Fig. S3B).**

L.366 The authors use the argument that knockout of S1P2 led to the inhibition of EGFR and as support for the inter-relatedness of this network. But in Fig 2D they show a profound decrease in GBS CFU presumably without involvement of EGFR, and of further downstream host factors. In many of the figures the concentration of the inhibitors or antibiotic are not clearly indicated. **As indicated above, our data indicates that GBS invasion of BBB depends on the S1P2-EGFR-CysLT1 network and counteracting any individual component was efficient in inhibiting GBS penetration into the brain and there was no additive effect by combining two components (Fig S4B). The concentrations of the inhibitors and antibiotics are provided**

Referee #3 (Remarks for Author):

In this manuscript, Zhu et al. report that GBS, the first etiologic agent for neonatal meningitis, exploits a signalling platform involving SphK1/2, S1P, S1P2 receptor, EGFR, PLA2 α , CysLT1 for penetration of the blood brain barrier by a transcellular mechanism. Blocking signalling molecules from this platform decreased GBS crossing into the brain in an in vivo model of meningitis and improved the outcome of the animals.

Major comments:

Molecular mechanisms for GBS penetration of the BBB are lacking. However, although results from this paper contribute to a better understanding of mechanisms used by GBS to cross the brain barrier, the novelty of the overall finding of this study is not so attractive for several important reasons:

1- SphK1/2, S1P, S1P2 receptor, EGFR, signalling pathway has already been described by the same group to be involved in BBB crossing by Escherichia coli K1, the other pathogen responsible for meningitis in neonates (Wang et al. PLoS pathogen 2016). More recently, the authors completed this signalling platform by the identification of PLA2 α and CysLTs downstream EGFR (Zhu et al. Cell Microbiol. 2020). Although it is very interesting that these 2 unrelated pathogens (Gram negative bacilli versus Gram positive coccus) use the same signalling pathway, identifying such pathway for another pathogen than E. coli K1 has a limited impact. The Wang et al. paper is referenced in the manuscript, but exclusively as a technical reference. Zhu et al. cell microbiol. 2020 paper is not referenced, the current paper seeming consequently more original than it is. The involvement of this signalling pathway that seems to be common for these 2 pathogens is not discussed in the manuscript. This point is of particular importance as it raises the question of GBS virulence factors involved in this platform activation. Indeed, in E. coli K1, OmpA, FimH and Nlp1 are the virulence factors responsible for S1P, EGFR, PLA2 α ,

CysLT1 activation (Wang et al. PLoS pathogen 2016) while those virulence factors are not expressed by GBS. Therefore, what are GBS virulence factors involved in the activation of this signalling pathway? – As indicated in response to reviewer 1, the mechanisms involved with GBS exploitation of S1P2-EGFR-CysLT1 differ from those with *E. coli* (Fig. S4A and S4C). In addition, determination of GBS factors contributing to invasion of the BBB remains incomplete, and we are waiting for such information to address this reviewer’s question.

Also, for *E. coli* K1, c-Src was identified as the signalling molecule between S1P2 and EGFR? Is it also the case for GBS? – c-Src was shown to be downstream of S1P2 and EGFR in GBS invasion of the BBB, as shown by the demonstration that c-Src phosphorylation in response to GBS was inhibited in EGFR and S1P2 KO cells, and this information is provided in Fig. S3F.

2- In addition to SphK1/2, S1P, S1P2 receptor, EGFR, signalling pathway, the authors further decipher the signalling cascade that also involves PLA2 α and CysLT1. As mentioned by the authors, the involvement of PLA2 α , CysLT1 in BBB crossing by GBS has been published previously by this group (Maruvada et al. IAI 2011). The involvement of CysLT1 in GBS invasion was demonstrated using Montelukast. It was also known that Montelukast inhibits GBS penetration of the BBB in vivo (Syu et al. Nature com. 2019). Because Fig. 3 B, C, D, E, F only confirmed previously published results, it should not appear in the results but as data not shown or supplementary information. - any figures identical to those of our previous publications (3B and E) were transferred to supplementary information. Our data in the CysLT2^{-/-} animals and CysLT1 knockdown HBMEC were not previously reported and were included in the text.

3- The very interesting point in the characterization of this signalling platform is that SphK1/2, S1P, S1P2 receptor, EGFR, signalling is connected to PLA2 α , CysLT1 signaling. However, data of this interconnection are not that convincing. The difference in phosphorylation status of PLA2 α in control versus EGFR KO (Fig. 3A right panel) or S1P2 KO (Fig. 3J) are very weak and the uninfected condition is not shown. The reviewer is not convinced by images of EGFR and ezrin recruitment around GBS associated cells (FIG 3L). – New Fig. 3B is provided to show the phosphorylation of cPLA2 α is inhibited in response to GBS (K79) in EGFR KO and S1P2 KO cells compared to uninfected controls.

In addition, new co-localization figures for K79 with EGFR with Ezrin are provided (Fig. 3K). To distinguish intracellular from extracellular bacteria, HBMEC were infected with GFP-K79 and counterstained in non-permeabilized condition with a rabbit anti-GBS serum. Then the cells were incubated with EGFR or Ezrin monoclonal antibody, and subsequently incubated with anti-mouse Alexa Fluor-488 and anti-rabbit Fluor-568 labeled secondary antibody. Under this condition, intracellular GBS were green, while extracellular bacteria were yellow. The co-localized bacteria with EGFR or ezrin were cyan (arrows).

4- Fig. 4 shows that Montelukast that acts as an antagonist of the CystLT1 receptor (at the end of the signalling pathway) in adjunct treatment with ceftriaxone strongly improves the outcome of infected mice. This result is particularly important and convincing. This part, that is central to the manuscript, should be further studied. Does gefitinib that act upstream also improves mice outcome? Are there any additional effects on survival if gefitinib and Montelukast are given together with ceftriaxone? – As indicated above, survival is also improved by the combination of ceftriaxone and gefitinib, but the outcome did not differ between combination therapy with single drug (gefitinib or montelukast) vs two drugs (Fig. S4B).

*p<0.05

5- Authors claim (lanes 122-124) that GBS penetration into the brain occurs in the meningeal and cortex capillaries by a transcellular mechanism. However, the data presented do not support this statement. No GFP bacteria are visible in fig 1B and Fig. S2 A, B and C (contrast problem or image format?). **We provided enlarged part for Fig. 2B and improved the contrast of Fig. S2.**

Hence, the reviewer cannot say that bacteria are visible outside of capillaries at early time points in cortex and meningeal capillaries while they are absent from choroid plexuses. In addition, the transcellular mechanism is not demonstrated. Indeed, authors claim it is a transcellular passage as bacteria can be found outside the capillaries while there was no extravasation of intravascular tracer (sulfo-NHS-Biotin). However, several groups fail to observe the diffusion of tracers during diapedesis of immune cells by paracellular mechanism at the BBB demonstrating that barrier function can be maintained during paracellular diapedesis of immune cells at the BBB (Winger et al. J Immunol 2014; Engelhardt et al. European Journal of immunology 2004...). The lack of tracer diffusion is therefore not sufficient to assert that GBS does not cross BBB by a paracellular mechanism.

Line 124. In order to further prove the structural integrity of the blood-brain barrier during the GBS penetration of the blood-brain barrier, claudin-5 staining was used to assess the tight junction at the blood-brain barrier endothelial cells. The claudin-5 staining one hour after GBS administration via the tail vein showed that GBS was shown in the brain capillaries (arrow) and successful penetration into the brain (arrowhead) without disrupting the barrier integrity (Fig. S3A). This finding is consistent with that of our functional integrity of the blood-brain barrier during GBS penetration of the blood-brain barrier, as shown in this report of no extravasation of intravascular small molecule tracer as well as no increased permeability of albumin (Kim et al 1997). These finding support GBS penetration of the blood-brain barrier via a transcellular mechanism.

The authors could assay GBS transcellular passage by performing transcellular migration assay on an appropriate cell line, such as HBMEC, in the presence of gefitinib or in EGFR KO cells. – penetration data across HBMEC monolayer on Transwell assays is provided (figures 2E and 3E)

Minor comments:

Fig. S2A time point 12hrs, capillary and GFP bacteria pictures have been inverted –corrected

Fig. 1A: some letters are truncated on the right of the figure. -revised

Lane 189: should it not be Fig. S3 instead of Fig. S2A? - corrected

Some experimental information is missing (origin of GFP-GBS strain? is GFP expression constitutive or inducible, chromosomal or plasmid encoded?) **constitutive and chromosomal**; which choroid plexus have been used in figure 1B and Fig. S2 C (lateral, 3rd, or 4th ventricle)?

The choroid plexus from 4th ventricle is shown here, but we showed the same findings in the choroid plexus from lateral ventricle.

Two different statistical tests described in material and methods (Wilcoxon and student) have been used for in vivo experiments. The reader should be informed of which test has been used for each figure. For comparison of paired bacterial counts from invasion assay, Student's t test was used. For the animal experiments with gefitinib and montelukast, two groups contain different numbers of mice, and we used Wilcoxon test to analysis the data. We added the information in the text. Also, for invasion experiments displaying more than 2 sets of data, the ANOVA test to correct for multiple comparisons would be more appropriate. – statistical methods are provided

Experiments have been performed in triplicate but authors should indicate the number of independent experiments that have been realized. – at least two or three independent experiments were done and this information was provided

11th Nov 2020

Dear Prof. Kim,

Thank you for the submission of your revised manuscript to EMBO Molecular Medicine. I am pleased to inform you that we will be able to accept your manuscript pending the following final amendments:

- 1) Please address the comments by the referee #2.
- 2) Figures: We noticed that some panels are presented in more than one figure (e.g. Fig 1B reused in Fig S2A and S2C and Fig 2C reused in Fig S2D). Please cross-reference all the reused panels in the figure legends.
- 3) In the main manuscript file, please do the following:
 - Correct/answer the track changes suggested by our data editors by working from the attached/uploaded document.

***** Reviewer's comments *****

Referee #1 (Comments on Novelty/Model System for Author):

The revised version answered to the reviewer's questions adequately and is considerably improved.

Referee #1 (Remarks for Author):

Impressive work!

Referee #2 (Comments on Novelty/Model System for Author):

No ethical issues. The authors did extensive experimentation to support their conclusions and added many new studies in response to the reviewers' comments. Given the observation that multiple interventions already approved for other indications were shown to be effective in this manuscript, follow up clinical studies may be warranted.

Referee #2 (Remarks for Author):

General comments:

Zhu et al did a thorough job in addressing the criticisms of each of the reviewers. They responded in detail to the issues raised, performed additional experiments and revised the manuscript accordingly. They discussed differences between K1-encapsulated E. coli and GBS, added Transwell assays to show that the GBS not only penetrated the HBMEC but also exited from the basal layer. They also showed specificity of the various inhibitors by showing these did not work in KO cells. There are a few issues unresolved. Using the cell-permeable inhibitor of clathrin-mediated endocytosis, dynasore, they show that GBS invasion of HBMEC involves clathrin-mediated endocytosis. They state that GBS is co-localized with EGFR after it is endocytosed (L. 298 and 981). Does the GBS bind to any of the cell surface receptors, such as EGFR or S1P-S1P2 receptor? Are there any known virulence factors possessed by GBS that induce the cascade of endocytosis? What GBS moiety, if any, activates EGFR? Why is such a high inoculum required for GBS invasion? While it is beyond the scope of this already inclusive manuscript, the investigators may want to comment in the discussion whether pericytes and astrocytes which ensheath the endothelial cells and are part of the blood brain barrier may also play a role in the GBS invasion. These cells express several pattern-recognition receptors that generate an immune response.

Specific Comments

L. 184: S1P levels significantly higher in HBMEC infected with GBS strain K79. I would assume that before GBS infection the S1P levels were not elevated, but that upon infection the S1P levels significantly increased. This raises the question as to whether after infection there is some positive feedback loop by which infection increases the level of the upstream S1P after it gets into the HBMEC which may facilitate the invasion of the yet-to-be engulfed GBS.

L. 302 not clear. Does the title imply that there are "counteracting host factors" that reduce GBS invasion, or is "counteracting" a verb that implies blocking host factors with specific inhibitors? Are S1P, EGFR and CysLT1 the "host factors" to which they refer?

The hypervirulent GBS type III clone, ST-17, accounts for up to 80% of meningitis cases. Is it known what virulence factor(s) they may possess that the other GBS strains do not? L. 553. Stationary phase bacteria were used in the GBS invasion assay. Do log phase bacteria behave differently? Fig S4D. The investigators show that Dynasore at 80 μ M blocks EGFR phosphorylation. What is the dynasore concentration in the middle lane represented only as a (+).

The authors performed the requested changes.

1) Answer for Referee #2

Referee #2 (Remarks for Author):

General comments:

There are a few issues unresolved. Using the cell-permeable inhibitor of clathrin-mediated endocytosis, dynasore, they show that GBS invasion of HBMEC involves clathrin-mediated endocytosis. They state that GBS is co-localized with EGFR after it is endocytosed (L. 298 and 981). Does the GBS bind to any of the cell surface receptors, such as EGFR or S1P-S1P2 receptor? Are there any known virulence factors possessed by GBS that induce the cascade or endocytosis? What GBS moiety, if any, activates EGFR? Why is such a high inoculum required for GBS invasion?

Based on our co-localization data, GBS is likely to bind to EGFR and S1P2.

As indicated in our previous response to the reviewer's comment, determination of GBS factors contributing to penetration of the blood-brain barrier remains incomplete as no genome-wide screen was not reported and we hope to address this reviewer's comment after comprehensive information of GBS factors is available. GBS penetration into the brain has been shown to be correlated with a high magnitude of bacteremia and a high inoculum required for GBS invasion is anticipated.

While it is beyond the scope of this already inclusive manuscript, the investigators may want to comment in the discussion whether pericytes and astrocytes which ensheath the endothelial cells and are part of the blood brain barrier may also play a role in the GBS invasion. These cells express several pattern-recognition receptors that generate an immune response.

We agree with the comment that this topic needs to be investigated in depth, but our previous studies have suggested that bacterial invasion of HBMEC did not differ between HBMEC with and without astrocyte co-cultivation.

L. 184: S1P levels significantly higher in HBMEC infected with GBS strain K79. I would assume that before GBS infection the S1P levels were not elevated, but that upon infection the S1P levels significantly increased. This raises the question as to whether after infection there is some positive feedback loop by which infection increases the level of the upstream S1P after it gets into the HBMEC which may facilitate the invasion of the yet-to-be engulfed GBS.

Our data suggested that activations of sphingosine kinases 1 and 2 are involved in increased S1P in HBMEC upon infection with GBS, but upstream molecules of S1P2 remain unclear at this time.

L. 302 not clear. Does the title imply that there are "counteracting host factors" that reduce GBS invasion, or is "counteracting" a verb that implies blocking host factors with specific inhibitors? Are S1P, EGFR and CysLT1 the "host factors" to which they refer?

Counteracting implies blockade and/or inhibition of host factors and this has been changed.

L 553. Stationary phase bacteria were used in the GBS invasion assay. Do log phase bacteria behave differently?

As shown in our previous studies (Nizet et al, 1997), GBS invasion frequency was similar between log-phase and stationary bacteria.

Fig S4D. The investigators show that Dynasore at 80 uM blocks EGFR phosphoryation. What is the dynasore concentration in the middle lane represented only as a (+).

It was a typo which should be (-). We corrected this mistake.

2) Figures: We noticed that some panels are presented in more than one figure (e.g. Fig 1B reused in Fig S2A and S2C and Fig 2C reused in Fig S2D). Please cross-reference all the reused panels in the figure legends.

We added the information in the figure legends.

11th Dec 2020

Dear Prof. Kim,

We are pleased to inform you that your manuscript is accepted for publication.

Corresponding Author Name: Kwang Sik Kim
Journal Submitted to: EMBO Molecular Medicine
Manuscript Number: EMM-2020-12651-V3